# Isthmin-1 (Ism1) modulates renal branching morphogenesis and mesenchyme condensation during early kidney development

Ge Gao[1,2,5], Xiaoping Li[3,5], Zhixin Jiang[2], Liliana Osorio[2], Ying Lam Tang[2], Xueqing Yu[1], Guoxiang Jin[1] & Zhongjun Zhou [1,2,4] ✉

The outgrowth of epithelial bud followed by reiterated bifurcations during renal development is driven by the ligand-receptor interactions between the epithelium and the surrounding mesenchyme. Here, by exploring ligand-receptor interactions in E10.5 and E11.5 kidneys by single cell RNA-seq, we find that Isthmin1 (Ism1), a secreted protein, resembles Gdnf expression and modulates kidney branching morphogenesis. Mice deficient for *Ism1* exhibit defective ureteric bud bifurcation and impaired metanephric mesenchyme condensation in E11.5 embryos, attributable to the compromised Gdnf/Ret signaling, ultimately leading to renal agenesis and hypoplasia/dysplasia. By HRP-induced proximity labelling, we further identify integrin α8β1 as a receptor of Ism1 in E11.5 kidney and demonstrate that Ism1 promoted cell-cell adhesion through interacting with Integrin α8β1, the receptor whose activation is responsible for Gdnf expression and mesenchyme condensation. Taken together, our work reveals Ism1 as a critical regulator of cell-cell interaction that modulates Gdnf/Ret signaling during early kidney development.

The kidney provides an efficient filtration and absorption function in removing wastes from the blood. Since the pioneering description of the morphogenesis and histochemistry of the developing mouse kidney[1], kidney development has been intensively investigated in both mice and humans. In higher vertebrates, kidney development undergoes three phases sequentially occurring in a rostral-to-caudal fashion, i.e., the pronephros, the mesonephros, and the metanephros which give rise to the definitive adult renal organ via the reiterative branching morphogenesis process of the ureteric epithelium and a MET-based (Mesenchymal-epithelial transition) nephrogenesis of the surrounding mesenchymal cells[2–4].

Branching morphogenesis occurs throughout the development of epithelial organs, including the lung, prostate, salivary gland, kidney, thyroid, and mammary gland[5]. It is driven by cell division, adhesion, migration, cell-cell communication and the surrounding extracellular matrix (ECM)[6,7]. It increases the contact surface of the epithelium for more efficient exchange or secretion, to achieve the optimal function. It is regulated spatiotemporally by consecutive ligand-receptor interactions between the epithelium and surrounding mesenchyme[8].

In kidney, the branching morphogenesis is a process starting from ureteric budding followed by reiterated bifurcations at the tips to form a specialized architecture. It is regulated by various transcription factors and growth factors including glial cell line derived neurotrophic factor (GDNF), fibroblast growth factors (FGFs), Wnt and transforming growth factor beta (TGF-β). Defective branching morphogenesis in kidney or urinary tract results in congenital anomalies of kidney and urinary tract (CAKUT), accounting for around 40–50% of chronic kidney disease in children. Renal agenesis (RA), an embryonic disorder

[1]Guangdong Cardiovascular Institute, Medical Research Institute, Guangdong Provincial People's Hospital, Guangdong Academy of Medical Sciences, Southern Medical University, Guangzhou 510080, China. [2]School of Biomedical Sciences, LKS Faculty of medicine, The University of Hong Kong, Hong Kong, China. [3]Department of Hepatic Surgery and Liver Transplantation Center of the Third Affiliated Hospital, Organ Transplantation Institute, Sun Yat-sen University, Guangzhou 510630 Guangdong, China. [4]Reproductive Medical Center, The University of Hong Kong - Shenzhen Hospital, Shenzhen, China. [5]These authors contributed equally: Ge Gao, Xiaoping Li. ✉e-mail: Zhongjun@hku.hk

referring to congenital kidney malformation with loss of one (unilateral RA) or both kidneys (bilateral RA) with a prevalence of 1:450-1:1000 births, belongs to one of the CAKUT categories[9]. So far, around 70 genes are reported relative to RA[10] and genes involved in renal agenesis are mostly regulators of Gdnf/Ret/Gfrα1 signaling, including transcription factors (Eya1, Six1, Pax2, Sall1, Hox11) and growth factors (Gdf11, Fgf, Npnt). Defects in cell-ECM adhesion also give rise to renal agenesis[11], during which Integrin signaling is particularly important.

Recently, single cell RNA-seq (scRNA-seq) was employed to depict detailed molecular changes during the spatiotemporal development in nephrogenesis in both mice (E12.5 to E16.5) and humans (wk7 to wk18)[12–14]. However, a systemic mapping of the first bifurcation event, the initial branching process before the nephrogenesis, is largely incomplete, especially on the mesenchyme-epithelium interaction. In this study, the communications between the different cell types were analyzed during the initial kidney branching morphogenesis (E10.5 and E11.5). Moreover, *Gdnf* co-expression analysis identified Isthmin-1 (Ism1) as a potential regulator of Gdnf/Ret signaling which was further demonstrated to be essential for kidney development.

Isthmin-1 (Ism1), initially identified in *Xenopus* as syn-expressed with Fgf8 at the midbrain-hindbrain boundary (MHB)[15], is a secreted protein characterized by two functional domains: a central thrombospondin type I repeats (TSR) domain[16] and a C-terminal adhesion-associated domain in MUC4 and other proteins (AMOP) domain[17]. The TSR domain is implicated in cell-cell or cell-ECM interactions, tumor metastasis, angiogenesis, TGFβ activation, would healing, and axon guidance[18,19]. The AMOP is an extracellular domain and is implicated in cell adhesion and tumor invasion[20]. The physiological functions of Ism1 remain largely unknown. Till now, Ism1 has been shown to inhibit tumor growth by suppressing angiogenesis[21,22]. Ism1 can increase vascular permeability[23] and is implicated in the hematopoiesis in zebrafish and craniofacial patterning in humans[24,25]. Ectopic Ism1 is reported to cause defective left-right asymmetry by negatively regulating Nodal signaling in chick embryos[26]. Recently, Ism1 is reported to serve as an adipokine to promote glucose uptake and improve hepatic steatosis[27]. In addition, Ism1 promotes GRP78-mediated alveolar macrophage apoptosis, thus serving as a potential therapeutic target in emphysema, a chronic obstructive pulmonary disease (COPD)[28].

In this work, we systemically explore the cell-cell crosstalk during the initiation of renal branching morphogenesis and identify *Ism1* as a potential gene responsible for CAKUT. We show that ISM1 promotes branching morphogenesis during early kidney development. Mice deficient for Ism1 fail to form the T-shape structure and condensed mesenchyme at E11.5. Gdnf/Ret signaling is significantly downregulated in the absence of Ism1. Additionally, by HRP-induced proximity-labelling, we identify Integrin α8β1 as the receptor of Ism1 in E11.5 kidney rudiments and reveal that Ism1 regulates mesenchyme condensation through Integrin α8β1-mediated cell adhesion. Thus, our work demonstrates a critical role for Ism1 during early kidney development.

## Results

### Ism1 was co-expressed with Gdnf in metanephric mesenchyme

To gain a comprehensive understanding of the branching morphogenesis during early kidney development, we employed 10x Genomics to profile 9852 cells and 11492 cells in E10.5 and E11.5 kidney rudiments, respectively. The kidney rudiment between E10. 5 and E11.5 is still not an independent tissue with clear margins, and in close proximity to adjacent tissues. Sequencing data covering two characterized stages (the budding and the first branching) during early kidney development, was time-resolved with a median of ~5000 genes/cell (Supplementary Fig. 1a–d). Based on the anatomical structure and canonical marker expression, we validated the identity of each of the cell clusters and categorized them into 12 clusters, including nephric progenitor

cells (NPC), stroma, nephric duct surrounding mesenchyme (NDM), surrounding mesenchyme (SurM), interstitial cells (ICs) and urothelial cells (Urothelium), ureteric epithelial cells (UE), urothelium and ureter epithelial cells (UrECs)[29], immune cells (Imm), blood, endothelial cells (Endo), and neural crest cells (Fig. 1b). The mesenchymal lineage was complex and consisted of Six2+ NPCs, Foxd1+ stromal cells, Epha4+/Tbx18+ NDM, Sfrp1+/Dcn+ SurM, Upk3d+/Podxl+ Urothelium, and Acta2+ ICs (Fig. 1b and Supplementary Fig. 1e).

To decipher the regulation of branching morphogenesis, UE and NPC lineage were further sub-clustered and the identity of each sub-cluster was assigned based on the expression profile. UE was divided into 4 sub-clusters, including nephric duct (ND), ureteric trunk (Trunk), ureteric bud cell or ureteric tip cells with high expression or low expression of *Ret* (Ret^High Bud&Tip and Ret^Low Bud&Tip) (Fig. 1c). NPC was divided into 3 sub-clusters, intermediate mesoderm (IntM), metanephric mesenchyme (MM) and cap mesenchyme (CapM) (Fig. 1d). Pseudotime analyses of UE and NPC, performed by Slingshot[30] were shown in Supplementary Fig. 2d, e.

Ligand-receptor crosstalk is of vital importance in regulating kidney branching morphogenesis, especially for the communications between UE and NPC lineages. Here, CellChat was employed to investigate the spatiotemporal crosstalk between sub-clusters of UE and NPC in branching morphogenesis, especially the ligand secreted from NPC subclusters. Cluster interactions were comprehensively analyzed based on a public repository of ligands and receptors. Amongst a variety of known signaling pathways that are involved in branching morphogenesis, GDNF was the most significant and specific ligand produced from CapM and received by Ret^High Bud&Tip cells (Fig. 1e). This confirmed the role for GDNF as a master regulator of branching morphogenesis, thereby providing the confidence to our unbiased ligand-receptor analysis.

To further explore the regulators whose receptors have not yet been established, secreted proteins in NPC cluster co-expressed with Gdnf were analyzed. Co-expression analysis has been used, in both bulk-RNA sequencing and scRNA-seq. To examine the protein-protein interactions (PPI) or to predict the functions of uncharacterized ligands[31–34]. Here, 8 ligands including *Sema3f, Vegfb, Fjx1, Sema3c, Ism1, Gdf11, Ntf3, Fgf9*, were identified in the top 20 ligands co-expressed with *Gdnf* in both E10.5 and E11.5 NPCs (Fig. 1f, Supplementary Fig. 3a, Supplementary Data 1 and 2). Among them, *Vegfb* was widely expressed in the whole mesenchyme lineage and *Fjx1* was mainly expressed in the UE1 lineage (Supplementary Fig. 3b). *Sema3c, Sema3f, Gdf11*, and *Fgf9* have been previously reported to positively regulate branching morphogenesis[35–38]. Ntf3/Ntrk2 has been shown to occur between NPC and cortical stroma during kidney development by crosstalk analysis[13]. Ism1 was the only unexplored ligand identified to be co-expressed with *Gdnf* (R^2 = 0.97 at E10.5 and R^2 = 0.73 at E11.5) during kidney development.

To verify the co-expression of *Ism1* and *Gdnf*, in situ hybridization and immunostaining were performed. At E10.75 and E11.25, *Ism1* was expressed in the mesenchyme surrounding the UB. Of note, *Ism1* was also found in the ND. At E11.5, *Ism1* mRNA was more prominent in the condensed mesenchyme. By E12.5, *Ism1* expression was lost in the epithelium and was expressed predominantly in the condensed mesenchyme (Fig. 1g).

### *Ism1* modulated renal branching morphogenesis in vivo and ex vivo

To validate the potential involvement of Ism1 in renal branching morphogenesis, *Ism1*-conditional knockout mice were generated by inserting the loxP-F3-Puro-IRES-GFP cassette into the intron 6 and loxP-Frt-Neo-loxP in intron 1. After removal of the selection genes, *Ism1*+/− mutant mice were generated by crossing with *Actb-Cre* mice to generate *Ism1*−/− mice. (Supplementary Fig. 4a). Deletion of *Ism1* gene led to GFP expression representing the endogenous *Ism1*. In E10.5

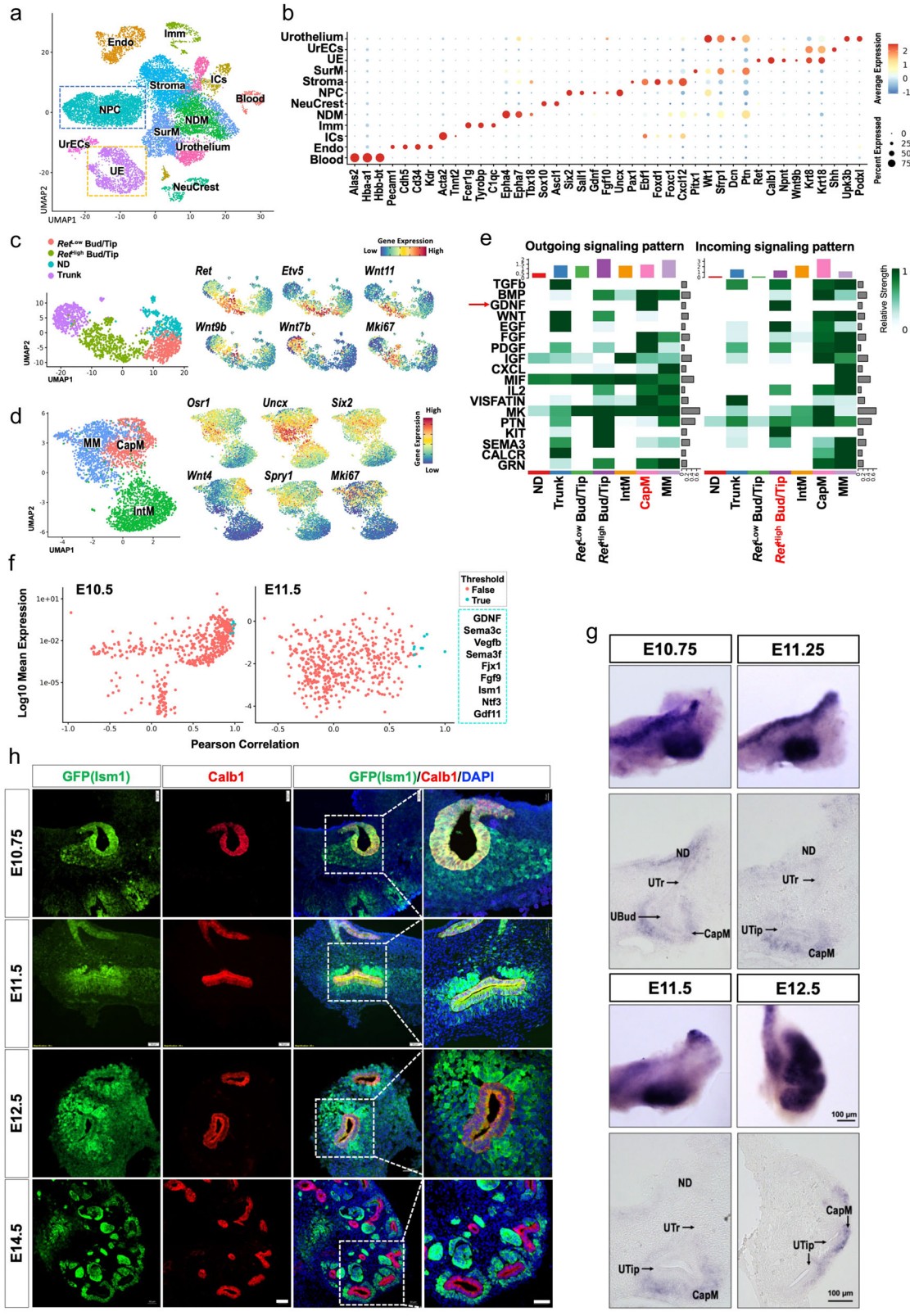

embryos, *Ism1* was predominantly expressed in MHB, the first and second branchial arch (BA1/2) and somites (Supplementary Fig. 4b). The *Ism1* heterozygous mutant mice were indistinguishable from their littermate wild-type controls.

Gross morphological examination of *Ism1⁻/⁻* mice showed significant kidney defects ranging from renal hypoplasia, dysplasia, unilateral renal agenesis (URA) to bilateral renal agenesis (BRA) (Fig. 2a and Supplementary Fig. 4d–f). No preference was observed either in

the asymmetry or between genders in the observed defects. *Ism1⁻/⁻* mice with BRA died within 24 h after birth. The urogenital tract was lost in either BRA or URA. However, no visible defects were observed in the adrenal gland, reproductive system, or bladder (Supplementary Fig. 4c). Based on the data from 170 *Ism1⁻/⁻* embryos collected between E14.5 and P0, approximately 60% of homozygous mutants exhibited various kidney defects, while such developmental abnormalities were observed in neither wild-type nor heterozygous mutant mice

**Fig. 1 | Identification of Ism1 as a candidate that co-expressed with Gdnf in the emergence of renal branching morphogenesis. a** UMAP plot showing 12 different cell types based on widely accepted markers in E10.5 and E11.5 kidneys. Cell type acronyms are shown in different colors. Endo endothelial cells, ICs interstitial cells, Imm immune cells, NDM nephric duct surrounding mesenchyme, NPC nephric progenitor cells, NeuCrest neural crest cells, SurM surrounding mesenchyme, UE ureteric epithelial cells, UrECs urothelium and ureter epithelial cells. **b** Dot plot depicting the expression of the marker genes in different cell types identified. **c** UMAP plots of different cell types in UE lineage and the distribution of individual marker gene in UE. **d** UMAP plots of different cell types in NPC lineage and the distribution of individual marker gene in NPC. **e** Crosstalk analysis between UE and NPC lineage within sub-clusters. Arrows indicate GDNF ligand which was secreted from CapM (red) and received by $Ret^{High}$ Bud/Tip (red). **f** Pearson correlation coefficient analysis of Gdnf in NPC lineage, candidates identified in both E10.5 and E11.5 samples are labelled as true (Pearson correlation >0.9 at E10.5 samples and >0.7 at E11.5 samples), listed in a green dotted box. Source data are provided as a Source Data file. **g** Examination of *Ism1* expression pattern by in situ hybridization at E10.75, E11.25, E11.5 and E12.5. Scale bar, 100 μm. Similar results are observed from at least 3 biologically independent experiments. ND nephric duct, UTr ureteric trunk, Ubud ureteric bud, UTip ureteric tip, CapM cap mesenchyme. **h** Examination of Ism1 expression by immunostaining from E10.75 to E14.5 in $Ism1^{+/-}$ mice. GFP represents endogenous *Ism1* expression. Scale bar, 50 μm. Similar results are observed from at least 3 biologically independent experiments.

(Fig. 2b). These results demonstrated that *Ism1* is critical for kidney development.

To understand why loss of *Ism1* gives rise to the kidney agenesis, we first examined its expression pattern in the developing kidney. $Ism1^{+/-}$ kidneys from E10.75 to E14.5 embryos were dissected and immunostained for GFP and Calbindin1 (Calb1), a marker for ureteric epithelium. Similar to the in situ results, *Ism1* was found in both UB and the surrounding mesenchyme, including metanephric mesenchyme (MM) and stroma along ND in E10.75 and E11.5 embryos. At E12.5, *Ism1* was enriched in condensed mesenchyme surrounding UB. At E14.5, *Ism1* was predominantly observed in the nephrongenic zone, including NPC and mesenchyme-derived structures including PTA, renal vesicle (RV), and S-shape bodies (Fig. 1h).

To understand how Ism1 regulates kidney development, we analyzed the kidney branching morphogenesis at E12.5 by whole-mount immunostaining of Cdh1. Notably, in $Ism1^{-/-}$ embryos, the ureteric bud failed to bifurcate, compared to that in the wild-type or heterozygous embryos that had already finished two cycles of branching (Supplementary Fig. 4g). In addition, immunostaining of cleaved-caspase3 at E11.75 showed the significantly increased apoptosis in mesenchyme where ureteric branching morphogenesis was absent, indicating defective interaction between the UB and mesenchyme in $Ism1^{-/-}$ embryos (Fig. 2c, d). Therefore, lack of *Ism1* compromised the kidney branching morphogenesis.

To further examine the developmental defect in renal branching morphogenesis, the kidney rudiments were dissected from E10.0 to E11.5 embryos and stained for Calb1. The UB induction and ND budding appeared normally in the $Ism1^{-/-}$ kidneys at E10.5. However, at E11.5, while the UB has invaded into the metanephric mesenchyme in the wild-type and bifurcated to form a T-shape structure, the first branching event was not observed in the $Ism1^{-/-}$ kidneys. The budding epithelium did not enter the mesenchyme despite the contact with mesenchyme (Fig. 2e, f). The condensation of the surrounding mesenchyme was also impaired, suggesting a compromised interaction between the mesenchyme and epithelium (Fig. 2g).

To dissect the defective renal branching morphogenesis observed in $Ism1^{-/-}$ kidney, explants of urogenital rudiments from E10.5 and E11.5 embryos and were taken into culture ex vivo on trans-well inserts. Consistent with the in vivo observations, wild-type explants exhibited a normal structure of the branching tree, while *Ism1* homozygous mutant explants from either E10.5 or E11.5 did not form a single T-shape structure (Fig. 2h, j). Therefore, renal agenesis or hypoplasia observed in $Ism1^{-/-}$ mice originated from defective branching morphogenesis, specifically from the first bifurcation.

To further demonstrate that the branching defect in $Ism1^{-/-}$ kidney rudiments was indeed a result from the loss of *Ism1* but not the GFP-fusion expression, both wild-type and *Ism1* mutant kidney rudiments were cultured on trans-well inserts in the presence or absence of recombinant Ism1 protein (rIsm1, 250 ng/ml). As shown in Fig. 2i, k, exogenous rIsm1 promoted UB branching in both wild-type and $Ism1^{-/-}$ explants, represented by more branching points and branching tips. The treatment with rIsm1, largely if not completely, restored the

defective UB branching in $Ism1^{-/-}$ explants from both E10.5 and E11.5 embryos (Fig. 2m, n). Taken together, defective UB branching morphogenesis was a direct consequence of *Ism1* deficiency.

## Difference in UE lineage between wild-type and $Ism1^{-/-}$ mice

To understand the mechanism behind the defective UB branching in $Ism1^{-/-}$ embryos, scRNA-seq was performed in 21344 wild-type cells and 16710 $Ism1^{-/-}$ cells, respectively, from kidney rudiments (Supplementary Fig. 5a, b). Concatenating the datasets from two genotypes at two time points revealed a clear segregation of transcriptionally divergent cell compartments. There was no significant difference in the cell ratio of each cluster between the two genotypes (Supplementary Fig. 5c). To further analyze the difference in main cell clusters between genotypes, UE and NPC lineages were segregated out and re-clustered.

To delineate the observed defects in ureteric bud bifurcation in $Ism1^{-/-}$ kidney rudiments, wild-type and $Ism1^{-/-}$ UE lineage cells captured from E11.5 were sub-clustered into ND&Trunk, $Ret^{High}$ Bud&Tip and $Ret^{Low}$ Bud&Tip (Fig. 3a), based on the expression profile (Supplementary Fig. 6a–c). Single-cell transcriptomes were analyzed in each sub-cluster from UE lineage. GO enrichment of DEGs from $Ism1^{-/-}$ $Ret^{High}$ Bud&Tip revealed the downregulation of genes in the regulation of cell motility, cell junction, cell adhesion, cell morphogenesis, cytoskeletal protein binding, actin-filament organization and cell adhesion molecule binding (Fig. 3b and Supplementary Data 3).

These data suggested that *Ism1*-null kidney rudiments were blocked in the ureteric bud cell state. In agreement, *Ret* ectopically accumulated in the *Ism1*-null ureteric 'bud-like' structure, a similar pattern was also observed in the co-receptor *Gfra*1 gene. However, *Wnt9b*, a marker for ureteric trunk, was upregulated at the ureteric "bud-like" structure at E10.5 in *Ism1*-null samples (Fig. 3c).

To validate the blockage of epithelial cell migration in the absence of Ism1 shown in Fig. 3b, CMUB-1 (Probetex, #W508) and CMMM-1 (#W501) cells were seeded in a trans-well inserts (Fig. 4k). CMUB-1 and CMMM-1 are immortalized cells lines derived from ureteric bud and metanephric mesenchyme at E11, respectively. The migration of epithelial cells was significantly reduced toward *Ism1*-deficient CMMM-1 (Supplementary Fig. 7a–c), compared to that toward wild-type CMMM-1. The addition of rIsm1 promoted the migration of CMUB-1 significantly, towards both wild-type and *Ism1*-deficient CMMM-1 cells (Fig. 3e, f), suggesting that Ism1 is a pro-migratory factor for ureteric epithelial cells.

## Difference in NPC lineage between wild-type and $Ism1^{-/-}$ mice

The *Six2*+ NPC lineage, the main population inducing ureteric epithelium branching, was further sub-clustered into CapM, MM and IntM (Fig. 4a). The identities of these sub-clusters were assigned based on the expression profiles (Supplementary Fig. 8a). To explore the differences in NPCs between wild-type and *Ism1*-null kidney rudiments, the proportion of each sub-cluster in NPC lineage was compared. The CapM percentage at E10.5 was significantly reduced in *Ism1*-null compared to their wild-type counterparts, suggesting the loss of nephric progenitors of mesenchymal origin (Supplementary Fig. 8b). In

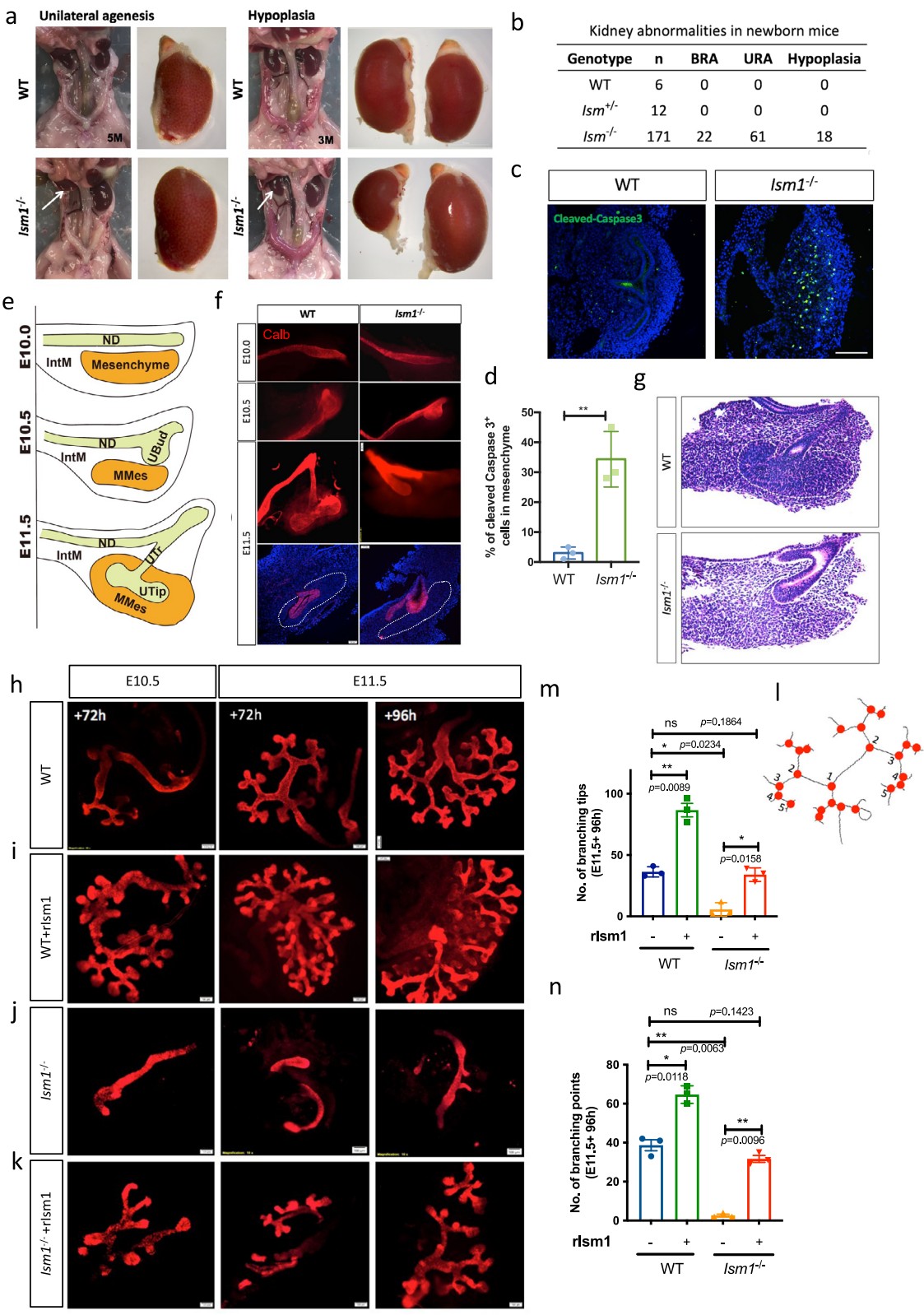

addition, the gene expression profiles in both MM and CapM at E10.5 revealed that transcription dysregulation occurred well before the visible defects in *Ism1⁻/⁻* kidney rudiments. DEGs (avg_log$_2$Fc > 0.25, $p < 0.001$) showed that loss of *Ism1* resulted in the downregulation of marker genes for cap mesenchyme or nephrogenic progenitors, such as *Sall1, Six2, Pax2* and *Itgα8* (Fig. 4c). GO analysis of the downregulated DEGs confirmed the impairment in cell junction, cytoskeleton, cell proliferation, tissue morphogenesis and actin binding in MM

(Supplementary Data 4), and urogenital system development, metanephros development, cell adhesion molecule binding in CapM (Fig. 4b and Supplementary Data 5). In situ hybridization and immunostaining further validated the downregulation of *Six2, Sall1 and* Integrin α8 (Fig. 4d–f). In wild-type kidneys, Integrin α8 exhibited a continuous staining at the boundary of the epithelium and mesenchyme, in contrast to the significantly decreased, and at times completely lost, signal in *Ism1⁻/⁻* kidneys. In addition, the ureteric bud surrounding

**Fig. 2 | Ism1 is required for renal branching morphogenesis. a** Renal agenesis, hypoplasia and dysplasia shown in adult *Ism1⁻/⁻* mice. **b** The percentage of abnormal kidney development in *Ism1⁻/⁻* mice from E14.5 to P0 (*N* = 171). URA unilateral renal agenesis, BRA bilateral renal agenesis. **c, d** Cleaved-caspase3 staining in E12.5 kidney rudiments to detect apoptosis in *Ism1⁻/⁻* mice (**c**) and quantification of Cleaved-caspase3 positive cells in mesenchyme of kidney rudiments from both WT and *Ism1⁻/⁻* embryos (**d**). Scale bar, 50 μm. Data are presented as mean ± SD from *n* = 9 (WT) and *n* = 9 (*Ism1⁻/⁻*). Data are from at least 3 biologically independent experiments as indicated. Statistical significance was determined using unpaired *t* test, two-tailed *p* = 0.0047. **e** Schematic diagram of the initiation and the first branching event during early kidney development from E10.0 to E11.5. ND nephric duct, IntM intermediate mesenchyme, UBud ureteric bud, MMes metanephric mesenchyme, UTr ureteric trunk, UTip ureteric tip. **f** Whole-mount immunostaining of Calbindin1 (epithelium marker) from E10.0 to E11.5 in both WT and *Ism1⁻/⁻* mice. **g** H&E staining

of E11.5 kidney sections from WT and *Ism1⁻/⁻* embryos, white dotted area indicates condensed mesenchyme in WT embryos which was absent in *Ism1⁻/⁻* embryos. **h–k** Calbindin1 staining of E10.5 and E11.5 kidney explants cultured for 72 h or 96 h in the presence or absence of rIsm1 at the concentration of 250 ng/ml. Scale bar, 100 μm. **l** Schematic diagram showing the skeleton of branching tree. Red point indicates the branching point. The number indicates the branching cycle. **m, n** Quantifications of branching tips and branching points per E11.5 kidney rudiment upon 72 h treatment of rIsm1, in both WT and *Ism1⁻/⁻* mice. Data are presented as mean ± SD from *n* = 18 (WT, 9 were cultured in the presence of rIsm1) and *n* = 18 (*Ism1⁻/⁻*, 9 were cultured in the presence of rIsm1). Data are from at least 3 biologically independent experiments as indicated. Statistical significance was determined using two-way ANOVA, and corrections for multiple comparisons were performed with Tukey (adjusted *p* values are indicated in the figure). Source data are provided as a Source Data file.

mesenchyme was disorganized and lost their orientation in *Ism1⁻/⁻* kidneys (Fig. 4d, e).

To validate the decreased mitotic cell cycle in *Ism1⁻/⁻* NPCs suggested in Fig. 4b, in vivo BrdU labelling (1 mg/10 g BW) was performed in pregnant mice. As shown by the co-staining of BrdU and Pax2 in Fig. 4g, the proliferation of mesenchyme in *Ism1⁻/⁻* embryos was significantly decreased compared to that in controls. However, proliferation in the ureteric epithelium was not affected in the absence of Ism1 (Fig. 4h, i). Taken together, loss of *Ism1* resulted in the decline in the nephric progenitors at a developmental point prior to branching morphogenesis, thereby resulting in the reduced proliferation in the NPC lineage.

### Loss of *Ism1* attenuated Gdnf/Ret signaling

Given that abnormalities were observed in both UE and NPC in *Ism1⁻/⁻* embryos, the crosstalk between these two lineages were examined by CellChat (Supplementary Fig. 9a, b). As shown in Fig. 5a, loss of *Ism1* resulted in reduced communication probability in UE/NPC crosstalk, particularly for the well-known signaling pathways for renal branching, Gdnf/Gfrα1 and Fgf10/Fgfr2 signaling.

To further validate the decreased Ret signaling in *Ism1⁻/⁻* kidney, the downstream effectors of the Gdnf/Ret signaling pathway were examined. As shown in Fig. 5b, the phosphorylation of Erk (pErk) and expression of transcription factor Etv5 were lost in UB cells in E11.5 *Ism1⁻/⁻* embryos. The attenuated Ret signaling was attributable to the transient loss of *Gdnf* expression at E11.5, as demonstrated by the whole-mounting in situ hybridization (Fig. 5c).

We then tested if exogenous rGDNF was able to restore Ret signaling and rescue the renal branching defects in *Ism1⁻/⁻* mice. As shown in Fig. 5d, both rGDNF and rIsm1 promoted the branching with an additive effect, suggesting that branching defects in the absence of *Ism1* are largely Gdnf/Ret signaling-dependent as GDNF could compensate for the loss of Ism1. This raised a possibility that Ism1 might regulate Gdnf expression. Indeed, *Gdnf* expression was increased at both transcriptional and protein levels upon rIsm1 treatment in CMMM-1 cells (Fig. 5e, f). The expression of *Gdnf* in primary mesenchyme isolated E11.5 *Ism1⁻/⁻* kidney rudiments were further examined, in the presence or absence of rIsm1 for 48 h. As shown in Fig. 5g, both *Gdnf* and *Six2* were upregulated in the presence of rIsm1. In addition, when E11 *Ism1⁻/⁻* kidney rudiments were taken into culture ex vivo in the presence of rIsm1 or rGDNF, Gdnf/Ret signaling was restored in the *Ism1⁻/⁻* kidney, as evidenced by immunostaining of its downstream targets, Etv5 and phosphorylation of Erk (Fig. 5h). Taken together, decreased expression of *Gdnf* in mesenchyme in the absence of *Ism1* jeopardized the Gdnf/Ret signaling, leading to defective renal branching morphogenesis in *Ism1⁻/⁻* mice.

### Ism1 enhanced Gdnf/Ret signaling via an "Integrin-like" pathway

To further elucidate the mechanism underlying GDNF regulation by Ism1, a modified peroxidase-mediated (HRP) proximity biotinylation

labelling was utilized to identify potential Ism1 receptors in E11.5 kidney rudiments (Fig. 6a). The proteome in the intact tissue, especially during the embryonic stage, is difficult to characterize by traditional methods. In addition, ligand-receptor interactions are normally intrinsically transient as a result of substrate turnover. Thus, proximity labelling (PL) is ideal in the situation to capture temporal and weak interactomes in situ[39–43]. Here, Flag-tagged Ism1 protein was conjugated with HRP with an efficiency over 70% (Fig. 6c). As shown in Supplementary Fig. 10a, N-terminal HRP conjugation did not hamper the effect of Ism1 on promoting renal branching morphogenesis in the ex vivo culture of E11.5 *Ism1⁻/⁻* kidney rudiments. HRP-mediated proximity biotinylation on Ism1 binding partners was performed with kidney rudiments followed by tissue lysis and Neutravidin enrichment. Eluted proteins were separated by SDS-PAGE followed by mass spectrometry (Fig. 6b). A wide range of proteins were detected as biotinylated Ism1 interacting partners in the embryonic kidney rudiments (Fig. 6d). However, only 3 membranous proteins were identified as potential Ism1 receptors, including Integrin β1, Plexin B2 and Ephrin B1 (Fig. 6e and Supplementary data 7). These three candidates were identified twice in three independent experiments.

Given that the TSR domain in Ism1 contain has a high affinity to Integrin β1[16,44–46], it is likely Ism1 is required for cell-cell adhesion through integrin signaling. Integrin β1 has also been shown to be necessary for the epithelial−mesenchymal interaction and to regulate organ branching morphogenesis[47,48]. In addition, though renal agenesis was observed in neither Integrin α3 nor α6 deficiency mutant mice[49,50], loss of Integrin α8 or its ligand Npnt both give rise to kidney agenesis with a penetrance similar to that in *Ism1* deficient mice[51,52]. Interestingly, defective Integrin α8β1 signaling also results in a transient downregulation of *Gdnf* at E11.5[53], a phenotype also observed in *Ism1* deficient mice. Therefore, it is not beyond our immediate speculation that Ism1 is a ligand for Integrin α8β1 and the defective mesenchyme condensation in *Ism1⁻/⁻* embryonic kidney may be a consequence of defective Integrin α8β1 signaling.

The interaction between Ism1 and Integrin β1 was confirmed by Co-IP in HEK293T where V5-tagged Integrin β1 (α8) and Flag-tagged Ism1 were co-transfected. As shown in Fig. 6f, g, both Integrin β1 and α8 could pull down Ism1. Additionally, both TSR and AMOP domain alone could be pulled down by Integrin β1. The endogenous interaction was further validated in situ by PLA assay. As shown in Fig. 6h, Ism1 binds to Integrin α8 at the junction of ureteric epithelium and the surrounding mesenchyme. The activation of Integrin α8β1 by Ism1 stimulation was also supported by the phosphorylation of downstream targets including FAK and Akt in CMMM-1 cells (Fig. 6j), and FAK, Akt and Erk in HEK293T cells with ectopic expression (Fig. 6k).

### Ism1 promoted cell-cell adhesion through Integrin α8β1

N-cadherin, a marker for cell-cell adhesion and enriched in mesenchyme surrounding the ureteric bud/tip, was significantly

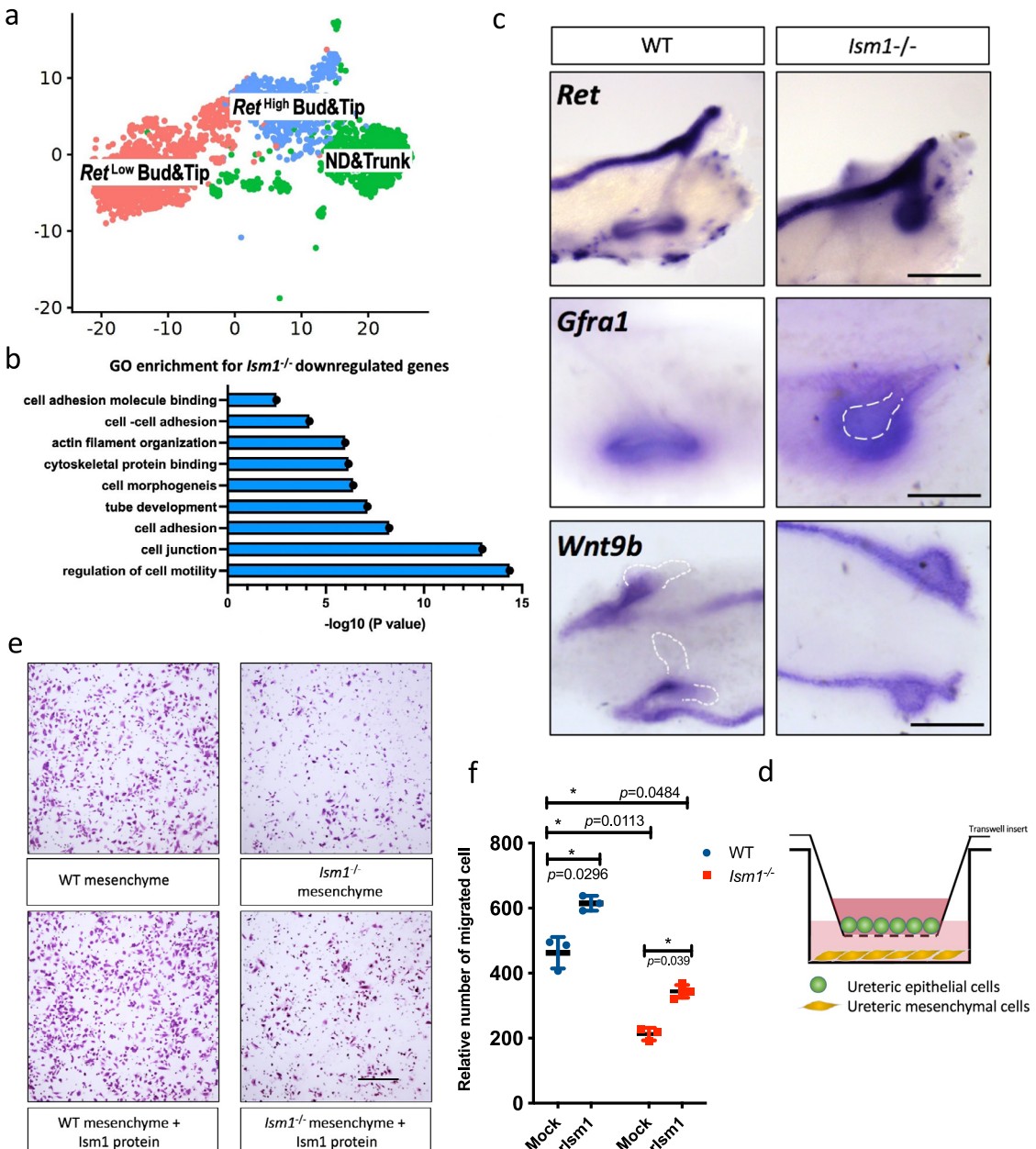

**Fig. 3 | Difference in UE lineage between wild-type and *Ism1⁻/⁻* mice during early kidney development. a** UMAP plots of different cell types in UE lineage from both WT and *Ism1⁻/⁻* mice. **b** GO enrichment of downregulated DEGs in *Ret*^High Bud&Tip sub-cluster in *Ism1⁻/⁻* kidney rudiment, as compared with WT. See Supplementary Data 3 for the full list of downregulated DEGs. **c** In situ hybridization of ureteric epithelial markers in E11.5 kidney rudiments, from both WT and *Ism1⁻/⁻* mice. Scale bars, 200 μm. **d** Diagram of the ureteric epithelial cell migration assay. Ureteric epithelial cells were plated in the upper well of a trans-well device. Mesenchymal cells, *Ism1*-null mesenchymal cells were plated in the lower wells in the presence or absence of rIsm1. **e–f** Crystal violet staining of ureteric epithelial cells migrated to the lower well (**e**) and quantification (**f**). Scale bar, 100 μm. Data are presented as mean ± SD from three independent experiments. Images of migrated CMUB-1 cells were obtained from five to eight random views per experiment. Statistical significance was determined using two-way ANOVA, and corrections for multiple comparisons were performed with Tukey (adjusted *p* values are indicated in the figure). Source data are provided as a Source Data file.

decreased in *Ism1⁻/⁻* embryonic kidneys (Fig. 7a and Supplementary Fig. 11b). In the hanging drop assay to resemble the in vivo mesenchyme condensation process, spheroid formation capability of CMMM-1 cells was significantly impaired with a loose morphology in the absence of rIsm1, compared to the compact structure in the presence of rIsm1 (250 ng/ml) (Fig. 7b, c).

To investigate the functional link between Ism1/integrin α8β1 and cell-cell adhesion, the aggregation assay was conducted in CMMM-1 cells with RGD peptide (the inhibitor of integrin α8β1) or AIIB2 (the integrin β1 inhibitory antibody) followed by rIsm1 (Ism1 conditioned medium) treatment. In the mock group, the number of cell aggregates increased (Fig. 7d, e) and N-cadherin expression was upregulated upon Ism1 or Npnt conditioned medium (as positive control) stimulation (Fig. 7f, g). Similarly, rIsm1 also increased the N-cadherin expression and enhanced the aggregation in HEK293T expressing Integrin α8β1 but not in HEK293T expressing integrin β1 only (Supplementary Fig. 11a), supporting the role of Ism1/Integrin α8β1 in promoting cell-cell adhesion. In addition, both RGD and AIIB2 pretreatment eliminated the Ism1-triggered upregulation of cell aggregates and N-cadherin expression, which was observed in both immunofluorescence and western blotting results (Fig. 7f, g), supporting functional evidences for Ism1/integrin signaling-

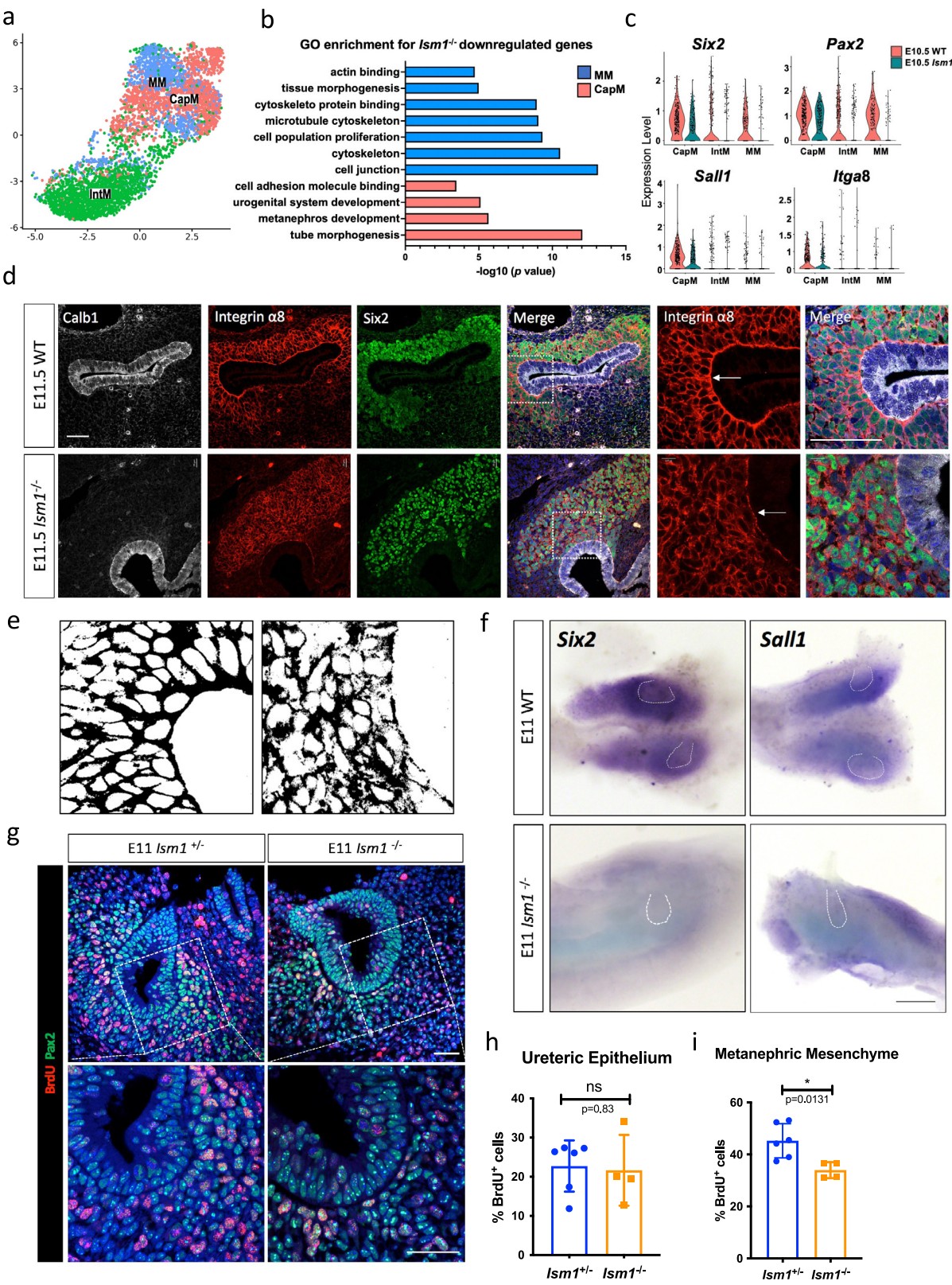

mediated cell aggregation or cell-cell adhesion. The phosphorylation of integrin signaling downstream targets FAK, AKT, and Erk was elevated upon Ism1 treatment, and was attenuated in the presence of RGD (Fig. 7g). Impaired Integrin signaling was also observed in E11.5 *Ism1⁻/⁻* kidney tissue as evidenced by the loss of phosphorylated FAK especially in the boundary between mesenchyme and epithelium (Fig. 7h). The role of Ism1 in activating Integrin signaling

was further validated in the mesenchymal cells isolated from E11.5 *Ism1⁻/⁻* kidney rudiments. The phosphorylated FAK and the expression of *Cdc42*, a downstream target of Integrin signaling, were both upregulated upon rIsm1 treatment. Taken together, these data demonstrated that Ism1 regulated mesenchyme condensation by promoting cell-cell adhesion through interacting with integrin α8β1(Fig. 7k).

**Fig. 4 | Difference in NPC lineage between WT and *Ism1*⁻/⁻ mice during early kidney development. a** UMAP plots of different cell types in NPC lineage from both WT and *Ism1*⁻/⁻ embryos. **b** GO enrichment of downregulated DEGs of CapM and MM sub-clusters in *Ism1*⁻/⁻ kidney compared with the WT at E10.5. See Supplementary Data 4 and 5 for the full list of downregulated DEGs in MM and CapM, respectively. **c** Violin plot showing the expression of the mesenchyme marker genes in sub-cluster of NPC lineage at E10.5. **d** Immunostaining of Itgα8, Six2 and Calb1 in WT and *Ism1*⁻/⁻ kidney rudiments at E11.5. Arrow indicated the boundary between epithelium and mesenchyme. Scale bar, 50 μm. The right panel showed a higher magnification of the areas indicated in the left panel. **e** The cell shape and orientation in UB-surrounding mesenchyme in E11.5 WT and *Ism1*⁻/⁻ kidney were analyzed

by ImageJ from Itgα8 staining. **f** Validation of downregulated DEGs by whole-mount in situ hybridization. *Six2* and *Sall1* expression in the kidney rudiments at E11 in wild-type and *Ism1*⁻/⁻ embryos. The white dotted lines highlighted the ureteric epithelium. Scale bar, 200 μm. **g** BrdU staining in sections of kidney rudiments from *Ism1*⁻/⁻ and littermate control embryos *Ism1*⁺/⁻ at E11. Higher magnification showed the region framed in white. Scale bars, 50 μm. **h, i** Quantifications of BrdU-positive cells in the ureteric epithelium (left) and metanephric mesenchyme (right) from *Ism1*⁻/⁻ (*n* = 4) and littermate controls (*n* = 6) at E11. Images were obtained from three to five random views per kidney rudiment. Statistical significance was determined using the unpaired *t*-test with the two-tailed *p* value as shown in figure, error bars are ± SD. Source data are provided as a Source Data file.

## Discussion

Kidney is among the most complex organs in terms of cellular composition and cell-cell communications[54]. In recent years, scRNA-seq has allowed a more detailed dissection of kidney organogenesis, particularly in the nephrogenesis process, in both humans and mice[55]. In current study, we systemically examined the early event of renal branching morphogenesis by scRNA-seq and explored the interactome between different lineages. By Gdnf co-expression analysis in NPC lineage, we demonstrated that Ism1 is a critical regulator for renal branching morphogenesis.

Mice deficient for *Ism1* failed to form the T-shape structure and condensed mesenchyme at E11.5. These defects have also been observed in embryos deficient for several extracellular ligands or corresponding receptors, including Gdnf, Npnt, Itgα8, Hs2st1, Fgf9/20, which suggests the potential involvement of Ism1 in these signaling pathways[38,51–53,56–59]. Indeed, Gdnf/Ret signaling was significantly downregulated in the absence of *Ism1*. The ex vivo culture showed that impaired branching in *Ism1*⁻/⁻ kidney rudiment was restored in the presence of Gdnf. In addition, *Gdnf* itself was transcriptionally downregulated in *Ism1*⁻/⁻ kidney rudiment whereas upregulated in CMMM-1 in response to Ism1 stimulation. Taken together, these results supported the involvement of Ism1 in Gdnf/Ret signaling during renal branching.

We also revealed that Ism1 regulates Gdnf/Ret signaling through integrin α8β1. By PL-MS in E11.5 kidney rudiments, Ism1 was shown to interact with Integrinβ1. In embryonic kidney, integrin β1 exists as α3β1 and α6β1 heterodimers in UE and α8β1 in the surrounding mesenchyme[47]. Integrin α3-null mice exhibit a reduced number of collecting ducts and abnormal glomerulus in kidney[50]. Selective deletion of integrin α3 in the UB results in abnormal or absent of the kidney papillae[60]. Conditional deletion of integrin β1 in the collecting system leads to a medullary defect[48]. Mice compound mutant for integrin α3 and α6 fail to develop ureters on both sides. However, renal agenesis was not reported in either integrin α3 or α6 deficient mice or compound mutant mice[49]. Hence, we speculate that there is a slim chance for integrin α3β1 or α6β1 to serve as the receptors for Ism1 during renal branching morphogenesis. The integrin α8β1 was further shown to interact with Ism1 and Ism1 activated integrin α8β1 signaling, demonstrated by the elevated phosphorylation of FAK, to promote cell-cell adhesion and mesenchyme condensation, evidenced by N-Cadherin expression and cell aggregation assay. Given that the deficiency in *Itga8* or its ligand *Npnt* give rise to similar defects in renal branching morphogenesis to that in *Ism1*⁻/⁻ mutant embryo, integrin α8β1 on mesenchyme likely serves as Ism1 receptor. Since integrin α8β1 signaling has been reported to regulate Gdnf expression and *Gdnf* expression was transiently lost in in E11.5 *Ism1*⁻/⁻ kidney, it is plausible that Ism1 regulates mesenchyme condensation and Gdnf/Ret signaling by activating Integrin α8β1.

The fact that in the absence of MM, Ism1 didn't significantly promote the branching of isolated UB cultured in gel (Supplementary Fig. 12a, b), suggested that either Ism1-mediated promotion of branching morphogenesis requires additional component(s) secreted from MM, or signaling pathway(s)through which Ism1 mediated branching have already been fully activated in the presence of FGF1,

GDNF, RA, R-Spondin1 in the culture medium. In line with these, recombination culture using isolated UB and MM from WT and *Ism1*⁻/⁻ embryos showed that the branching defect in *Ism1*-null UB was rescued either by co-culture with WT MM or by exogenous rIsm-1 when co-cultured with *Ism1*⁻/⁻ MM. In addition, the WT UB recapitulated the branching defect when co-cultured with *Ism1*-null MM and was restored when exogenous rIsm1 was present (Supplementary Fig. 12c−e). Taken together our data suggested that Ism1 promotes renal branching morphogenesis in a MM-dependent manner.

Notably, Ism1 deficiency gave rise to kidney agenesis or dysplasia with at least 60% penetrance from E14.5 to P0 in homozygous KO mice in a C57BL/6 background. However, the penetration was even higher (more than 80%) when examined at E11.5. As illustrated in Supplementary Fig. 13a, five different types of UB morphology could be observed in the mutant embryos at E11.5. Type I represents normal UB branching which account for about 20%. Among the UB without branching structure, 20% were type II UB which was also appeared in wild-type and heterozygous embryos, though with lower percentages (11% and 15%, respectively). Type II represents those with delayed development and likely will undergo normal branching morphogenesis. These percentage coincides well with the observed percentage of defective branching morphogenesis at E11.5 (>80%) and defective kidney development at birth (59%).

It is worthy to note that despite known as a ligand for Integrin α8β1, Npnt is restricted to the boundary between the epithelium and mesenchyme in the kidney rudiment. How Integrin α8β1 is activated beyond the boundary especially in the mesenchyme remains unknown. It is likely that in the mesenchyme, Ism1 serves as the ligand for Integrin α8β1. The biological relevance of Ism1 in Integrin and Gdnf signaling revealed in this study is not necessarily limited to the kidney development. Ism1 may also be involved in the development and functions in other organs, such as in brain and reproduction system where Ism1, Integrin and Gdnf/Ret signaling are all prominently expressed. In conclusion, we identified Ism1 as a candidate disease gene for CAKUT and provided a mechanistic explanation underlying the congenital renal defects in the absence of Ism1 in mice. Our findings provide insights into the regulation of Integrin α8β1 and Gdnf/Ret signaling during early kidney development.

## Methods
### Mouse models
Animal work was performed under the permission of the Committee on the Use of Live Animals for Teaching and Research (CULATR) following the guidelines and regulations. All mice were on a C57BL/6 J background raised in the normal 12/12 light-dark cycle. *Actb-Cre* males were crossed with 8-week-old *Ism1*-floxed female mice to generate heterozygous and homozygous *Ism1*-CKO;*Actb-Cre* mice. Both homozygous and heterozygous *Ism1*-CKO;*Actb-Cre* mice were normal in reproduction system, thus they were used for obtaining *Ism1*⁺/⁻ or *Ism*⁻/⁻ embryo. The noon of the day on which a vaginal plug was found was designated as E0.5. To obtain embryo samples from the different stages, the pregnant females were sacrificed by cervical dislocation, and the embryos were dissected in PBS buffer on ice.

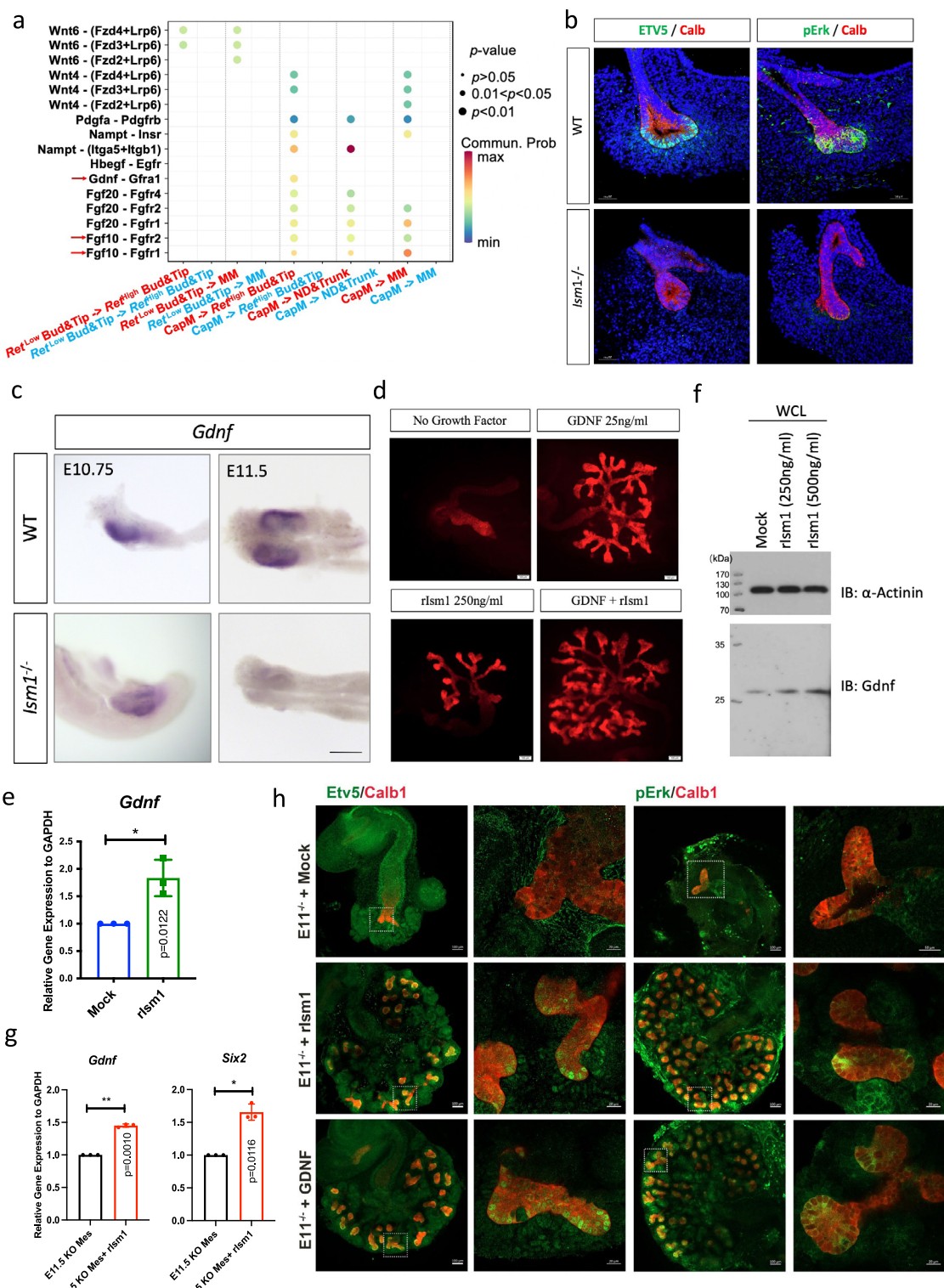

**Fig. 5 | Attenuated Gdnf/Ret signaling in *Ism1*$^{-/-}$ mice. a** Comparison of crosstalk between UE and NPC lineage in WT and *Ism1*$^{-/-}$ kidney rudiments. Dot color represents communication probabilities and dot size represents computed *p*-value. Empty space indicates that the communication probability is zero. *p*-values are computed from one-sided permutation test. **b** E11.5 kidney sections from WT (top) and *Ism1*$^{-/-}$ mice (bottom) were immunostained with Calb1 (Red) and Etv5 (Green) or pErk (Green). Scale bars, 50 μm. **c** In situ hybridization of *Gdnf* in E10.75 and E11.5 kidney rudiments from *Ism1*$^{-/-}$ and WT mice. Scale bar, 200 μm. **d** Calbindin1 staining of Kidney rudiments explants from E11.5 *Ism1*$^{-/-}$ embryos cultured for 72 h in the presence or absence of Gdnf (25 ng/ml) and rIsm1 (250 ng/

ml). Scale bars, 100 μm. **e, f** Gdnf expression detected by qPCR (**e**) and Western Blotting (**f**) upon Ism1 treatment (48 h) in CMMM-1 cell line. WCL, whole cell lysis. **g** Expression of *Gdnf* and *Six2* detected by qPCR upon rIsm1 treatment for 48 h in primary mesenchymal cells isolated from E11.5 *Ism1*$^{-/-}$ kidney rudiments. Data (**e, g**) were obtained from 3 biologically independent experiments; error bars are ± SD. Statistical significance was determined using the unpaired *t*-test with two-tailed *p* value as shown in figure. Source data are provided as a Source Data file. **h** Immunostaining of Calb1 (Red) and Etv5 (Green) or pErk (Green) in E11 *Ism1*$^{-/-}$ kidney explants in the presence of rIsm1 or GDNF for 96 h. Scale bars, 100 μm or 20 μm as labeled.

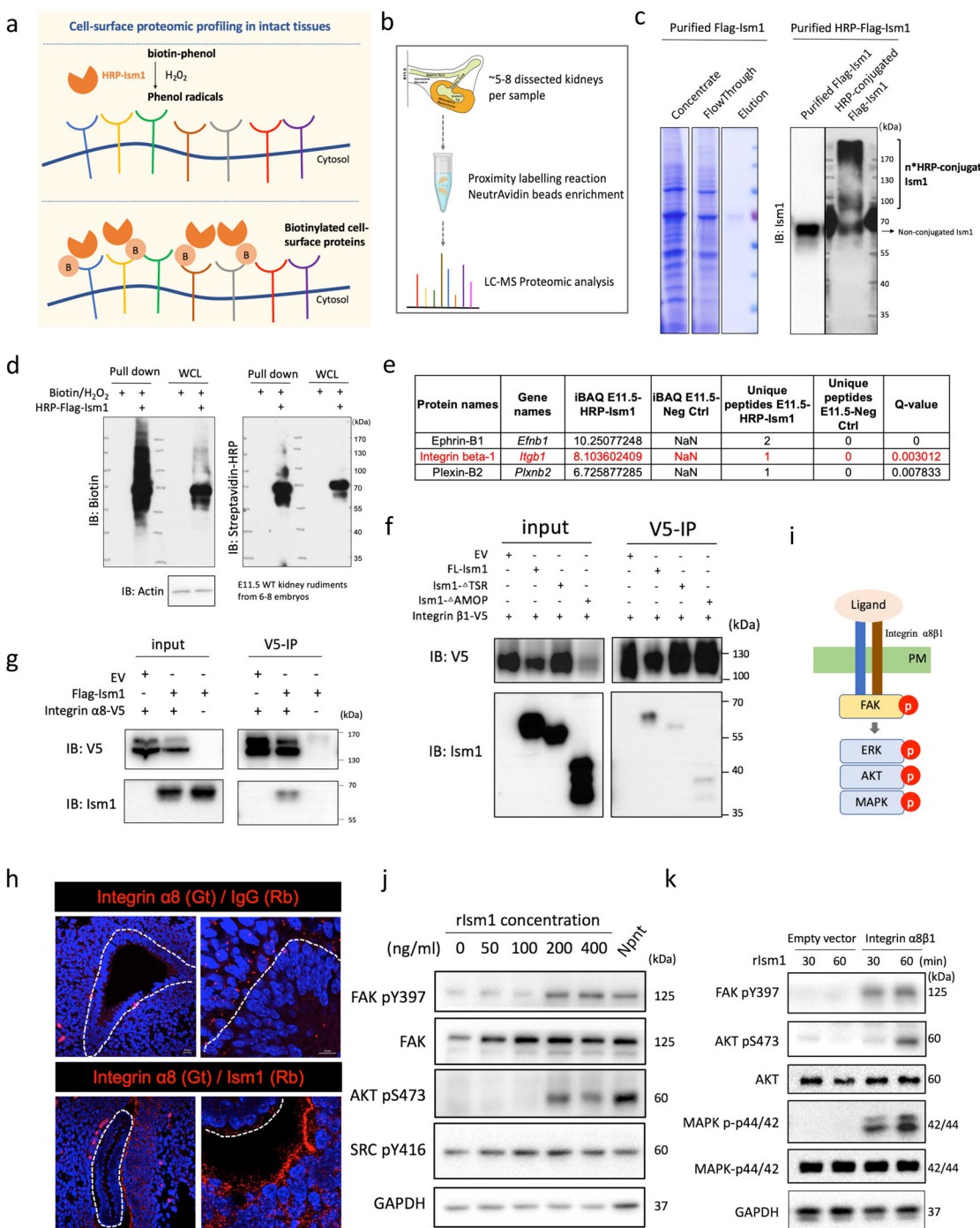

**Fig. 6 | Ism1 enhances Gdnf/Ret signaling through a potent "Integrin-like" pathway. a** Proximity labelling of peroxidase-based approach. Biotin-phenol was oxidized into reactive phenoxyl radicals with hydrogen peroxide existence, thus enabling the proximal labelling. **b** Proteomic workflow for mapping protein-protein interaction. Proximity labelling enzyme fused to Ism1 and reacts in E11.5 kidney tissues. Biotinylated proteins are enriched and analyzed by LC-MS. **c** Production and purification of HRP-conjugated Flag-Ism1 protein. Coomassie blue staining indicated the purified Flag-Ism1 protein from concentrated conditioned medium. Western blotting stained for Ism1 indicated the HRP-conjugated Flag-Ism1 and the efficiency of HRP conjugation is approximately 70% (the size of HPR-conjugated protein shifted from 70 kDa to 110 kDa and even over 170 kDa). **d** Biotinylated proteins from a proximity labeling experiment analyzed by biotin and streptavidin-HRP blot in E11.5 wild-type kidney rudiments. Clear changes in band pattern were observed for HRP-Flag-Ism1 treated samples compared with ligand-free samples. WCL, whole cell lysis. **e** Potential receptors identified by proximity labelling with mass spectrometry analysis. Summary of the mass spectrometry analysis was

provided in Supplementary Data 7. **f** HEK293T cells overexpressed with Integrin β1 and Ism1 truncation. Co-IP showed that the Integrin β1 cannot pull down full-length Ism1, TSR-deleted Ism1 and AMOP-deleted Ism1. **g** HEK293T cells overexpressed with Integrin α8 and Ism1 truncation. Co-IP showed that the Integrin α8 can pull down full-length Ism1 and AMOP-deleted Ism1. **h** Interaction between Integrin α8 and Ism1 were confirmed in situ by PLA with E11.5 wild-type kidney samples. Negative control was prepared with antibodies of Integrin α8 and IgG (Rb). **i** Model of canonical Integrin signaling. Upon the ligand binding to the integrin hetero-dimer, the signaling was activated by triggering phosphorylation of FAK, followed by phosphorylation of Erk, Akt and Mapk. PM, plasma membrane. **j** HEK293T cells transiently transfected with Integrin α8β1 were treated with rIsm1 for the indicated concentration and Npnt protein for 15 min. Integrin signaling downstream phosphorylation was detected by Western Blotting. **k** HEK293T cells transiently transfected with empty vector or Integrin α8β1 were treated with rIsm1 for the indicated time points. Integrin signaling downstream phosphorylation was detected by Western Blotting.

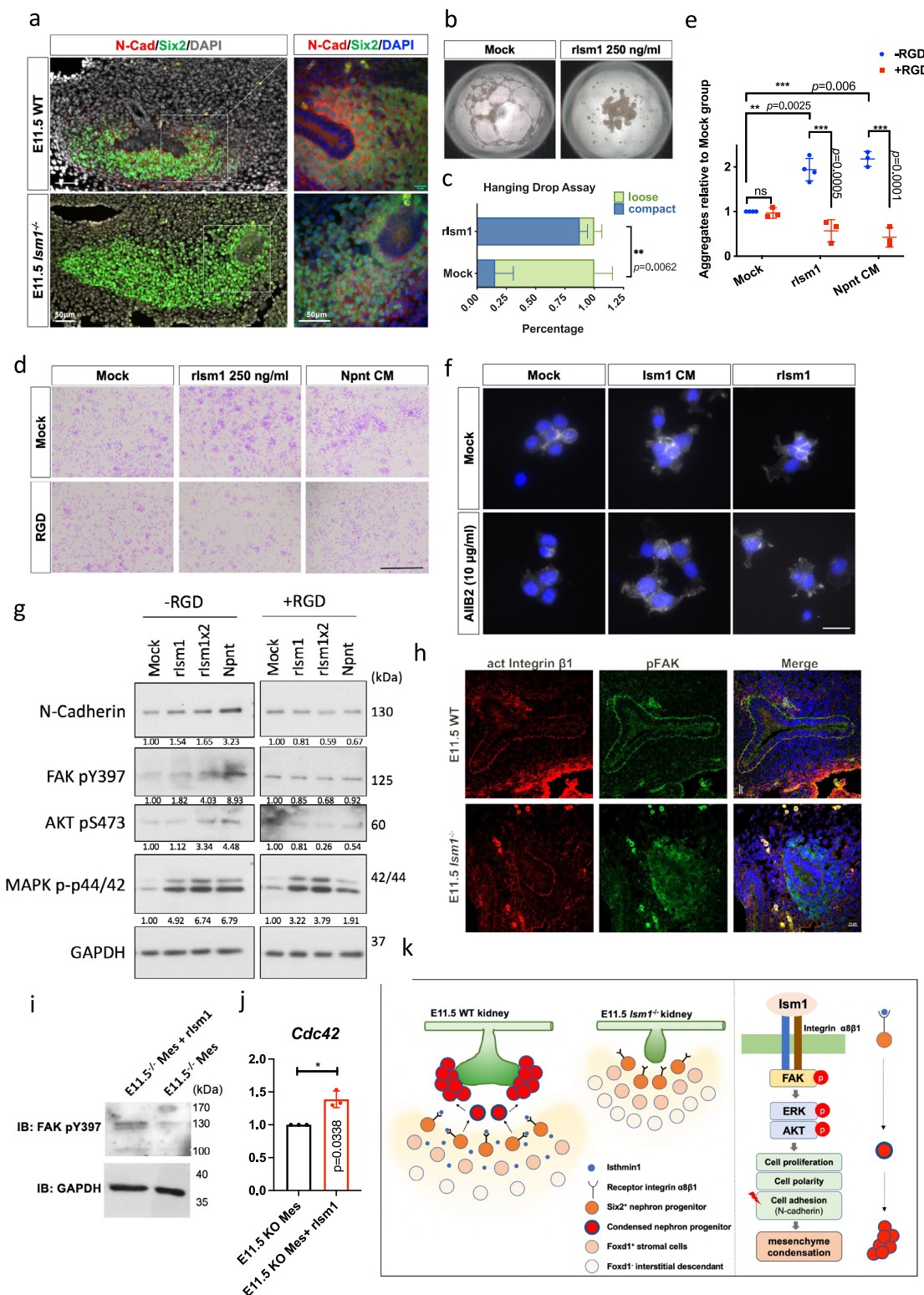

## Cell lines

Mouse metanephric mesenchyme (CMMM-1, #W501) and mouse ureteric bud (CMUB-1, #W508) cells obtained from Probetex, were maintained in DMEM (high glucose) medium with 10% FBS. *Ism1*$^{-/-}$ CMMM-1 cells were generated by CRISPR/Cas9. Cells were electroporated with Cas9 plasmids together with Dual gRNAs targeting exon 2 and exon 6 to delete both functional domains with LONZA reagent (VPI-1005; VPH-5012) using Nucleofector™ 2b Device followed by puromycin (1 μg/ml) selection. Single colonies were picked out and

transferred to 24-well plate. Genotyping PCR was done by primers flanking the two gRNA targeted sites, and confirmed by Sanger sequencing and Western blotting. Related primer information was provided in the Supplementary Data 6.

## Ex vivo culture of embryonic kidney rudiments

Embryo dissection was performed under a stereomicroscope. A 3 cm stainless-steel mesh was put into a 6-well plate and covered with 1.2 ml medium. DME/F12 was used as basic medium for kidney rudiments

**Fig. 7 | Ism1 promotes mesenchyme cell aggregation through Integrin.**
**a** Immunostaining of N-cadherin and Six2 in E11.5 kidney sections from WT or
*Ism1*⁻/⁻ mice. Left panel scale bars, 50 µm. Right panel scale bars, 20 µm. **b**, **c** Bright
field of hanging drop assay of CMMM-1 cells in the presence or absence of rIsm1 for
24 h (**b**). Scale bars, 200 µm. Quantitative data are shown according to the bright
field of hanging drop assay (**c**). Quantification of (**b**). Data from 3 independent
experiments and each group contains at least 20 drops for analysis. **d** Crystal violet
staining of aggregation assay in CMMM-1 cells. For the aggregation assay, CMMM-1
cells were pretreated with RGD peptide or saline and then subjected to rIsm1 or
Npnt CM. CM conditioned medium. **e** Quantification of (**d**). Data from 3 indepen-
dent experiments. Data are presented as mean ± SD from three independent
experiments. Statistical significance was determined using two-way ANOVA, and
corrections for multiple comparisons were performed with Sidak (adjusted *p* values
are indicated in the figure). Source data are provided in Source Data file.

**f** Immunostaining of N-cadherin in the aggregation assay of CMMM-1 cells. CMMM-
1 cells treated with either Ism1 conditioned medium (CM) or rIsm1, in the presence
or absence of AIIB2, the blocking antibody of Integrin β1. **g** CMMM-1 cells were
serum starved with FreeStyle293 medium for 4 h followed by pretreatment with
RGD peptide or saline. Cells were treated with either rIsm1 or Npnt conditioned
medium for 30 min followed by Western blotting to detect N-cadherin protein and
the phosphorylation of integrin downstream targets. **h** Immunostaining of active
form of Integrin β1 and pFAK in E11.5 kidney sections from WT or *Ism1*⁻/⁻ mice. Scale
bars, 20 µm. **i**, **j** Phosphorylation of FAK at Y397 detected by Western Blotting (**i**)
and *Cdc42* expression detected by qPCR (**j**) in primary mesenchymal cells isolated
from E11.5 *Ism1*⁻/⁻ kidney rudiments, in the presence or absence of rIsm1 for 48 h.
**k** Schematic model for Ism1-mediated Integrin regulation of cell-cell adhesion in
renal mesenchyme.

culture supplemented with 10% FBS, 100x penicillin-streptomycin and
GlutaMAX. The transwell insert (1.0 µm pore size) on the mesh was
prepared to form a liquid-air culture condition for embryonic kidney.
Kidney rudiments were dissected on ice from the urogenital system of
embryos and transferred onto the transwell inserts using a 10 µL pip-
ette. The culture plates were incubated at 37 °C with 5% CO$_2$ for
72–96 h. The medium was changed every 48 h.

### In vitro culture of isolated ureteric buds in the presence or absence of isolated mesenchymal cells
Embryonic UB was isolated and cultured as previously described[61,62].
Briefly, kidney rudiments (E10.5 or E11.5) were dissected and incubated
with Col IV as well as DNase I for 20–30 min at 37 °C. The digestion was
stopped by the addition of FBS. UB was dissected from surrounding
mesenchyme under stereomicroscope and transferred into Matrigel in
24-well inserts. The isolated UB in the gel was cultured with basal
medium (DMEM + 10%FBS + 1%P/S), supplemented with the following
growth factors, GDNF (1 ng/ml), FGF1 (100 ng/ml), RSPO1 (50 ng/ml)
and RA (1 mM). Culture medium was changed every 48 h.

For recombination experiments, the isolated mesenchyme was
dissected from UB and further digested into single cell suspension with
0.1% Trypsin for 10 min at 37 °C. The mesenchymal cells were seeded in
the lower 12-well plate with basal medium (DMEM + 10%FBS + 1%P/S)
while isolated UB was incubated in Matrigel in the upper inserts of 12
well plate. Medium was changed every 48 h. The growth and branching
of the isolated UB were examined and photographed after 4–8 days.

### Production and purification of Ism1 protein from HEK293T cells
Flag-tagged recombinant Ism1 was generated by expressing pFLAG-
CMV1-Ism1 plasmid in HEK293T cells. The pFlag-CMV1 plasmid con-
tains a preprotrypsin (PPT) fragment under the CMV promoter to
induce secretion of the target gene. After switching the medium to
DMEM supplemented with GlutaMAX, secreted Ism1 protein was col-
lected from the medium and concentrated by Amicon Filters Cen-
trifuge (30KDa molecular weight cut-off, MWCO) at 4000 g for 30 min
at 4 °C, followed by overnight incubation with the pre-washed Flag M2
magnetic beads at 4 °C with agitation. RIPA buffer was used to remove
unbound proteins before the Flag-Ism1 protein was competitively
eluted in Flag peptide (1 mg/ml) buffer by shaking for 16 h at 4 °C. After
dialysis was performed to remove the Flag peptide, the concentration
of the purified Flag-Ism1 was determined by the DC Protein Assay (Bio-
Rad) followed by Coomassie Blue staining.

### Protein extraction from cells or tissues and Western blotting
Briefly, cells were harvested and lysed in RIPA lysis buffer containing
protease inhibitor cocktail (11836145001, Roche) and phosphatase
inhibitor cocktail (78420, Thermo Scientific). Cleared samples with 6x
SDS loading buffer were denatured by boiling and resolved by SDS-
PAGE. Similarly, tissue samples were first minced thoroughly and then
electronically homogenized in RIPA buffer followed by 2 rounds of

sonication (25% amplitude/power, 10 s) before centrifuge and SDS-
PAGE separation. Antibodies used for Western blotting, Co-IP and
Integrin signaling blocking include mouse anti-ACTB (1:5000, A5316,
Sigma), rabbit anti-GAPDH (1:5000, 10494-1-AP, Proteintech), mouse
anti-FLAG (1:2000, F1804, Sigma), rabbit anti-Ism1 (1:5000, Genescript),
mouse anti-V5 (1:3000, R960-25, Invitrogen), rabbit anti-
phosphorylated FAK (1:1000, #3283, CST), rabbit anti-phosphorylated
AKT (1:1000, #9271, CST), rabbit anti-phosphorylated ERK (1:1000,
#4370, CST), rabbit anti-phosphorylated SRC (1:1000, #2101, CST),
mouse anti-N-cadherin (1:2000, #610920, BD), Goat anti-Gdnf (1:2000,
AF212, R&D), HRP-conjugated mouse (1:10000, GE Healthcare, NA9310)
and HRP-conjugated rabbit (1:10000, GE Healthcare, NA9340).

### Real-time fluorescence quantitative PCR (RT-qPCR)
Total RNA was isolated from CMMM-1 cells or tissues by TRIzol reagent
(RNAiso Plus, Takara Bio, #9109) according to the manufacturer's pro-
tocol. RNA was reverse transcribed using High Capacity cDNA Reverse
Transcription Kit (Thermo Fisher Scientific, #4374966) according to the
manufacturer's instructions. The real-time PCR was performed using TB
Green Premix Ex Taq (Takara Bio, RR420A) and data were normalized to
*Gapdh*. The primers used are listed in Supplementary Data 6.

### Co-immunoprecipitation
Proteins were prepared from HEK293T cells transiently transfected
with V5-tagged Integrin α8 (or Integrin β1) and Flag-tagged Ism1 or
Ism1 truncation plasmids, with a sonication step to release proteins on
chromatin. The supernatant was collected, with a small portion as
input, to incubate with appropriate antibodies at 4 °C for 4–6 h before
mixing with Protein A- or G-agarose beads overnight. The protein/
antibodies/ beads complex were eluted in 2xSDS loading buffer by
boiling and analyzed by Western blotting.

### Growth factor stimulation
HEK293T cells transfected with Integrin α8β1 were seeded in 12 or 24-
well plates and switched to FreeStyle293 Expression medium for
serum starvation for 4 h at 70% confluence, followed by rIsm1 treat-
ment. For CMMM-1 cells, starvation was for 2 h at 80% confluence.

To block the integrin signaling, cells were pre-treated with RGD
peptide or antagonistic antibodies AIIB2 (1 µg/ml) against integrin β1
for 30 min before rIsm1 stimulation. Cells were then washed with cold
PBS followed by protein extraction and Western blotting.

### Identification of Ism1 receptor using HRP-mediated proximity labelling followed by mass spectrometry
**Production and purification of HRP-conjugated Ism1.** Flag-Ism1
protein was conjugated with HRP using the EZ-Link Plus Activated
Peroxidase kit (#31489, Thermo Scientific) according to the protocol
provided by the manufacturer. Briefly, Flag-Ism1 was conjugated with
activated peroxidase (1:1) in Carbonate-Bicarbonate buffer (pH 9.4).
After removing the Non-conjugated HRP, proteins were washed with

RIPA buffer at least five times and eluted with 1 mg/ml Flag-peptide by shaking at 4 °C for 4–6 h, followed by concentration with the Amicon Filters Centrifuge (30 MWCO) to remove residual Flag peptide. Purified HRP-conjugated-Flag-Ism1 was checked by Coomassie Blue staining.

**HRP-mediated proximity labeling on embryonic kidney rudiments.** E11-E11.5 embryonic kidneys were dissected in pre-cooled PBS. Kidney rudiments from each embryo were randomly prepared in two Eppendorf tubes as control group and experimental group, respectively. Kidney rudiments were pre-incubated with 500 µM biotin-phenol (diluted in PBS, purchased from MCE) on ice for 30 min, followed by rIsm1 protein (5 µg/ml, as the control group) or HRP-Ism1 protein (5 µg/ml, as the experimental group) stimulation for 5 min on ice, with occasional mixing via pipetting. Hydrogen peroxide was added at a final concentration of 1 mM to initiate the labelling process for 1 min, followed by washes for 5 times with quenching buffer (10 mM ascorbate acid, 5 mM Trolox, 10 mM sodium azide in DPBS). For biochemistry detection and proteomic identification, kidney rudiments were snapped frozen in liquid nitrogen and transferred to the lysis buffer.

**Enrichment of biotinylated proteins.** RIPA buffer was added to frozen kidney rudiments to lyse the tissue followed by two rounds of sonication (25%, for 10 sec). After centrifugation for 30 min at 13,000 x g at 4 °C, the supernatant was collected and incubated with 75 µL NeutrAvidin beads O/N with gentle rotation at 4 °C to pull down the proximal proteins of HRP-Ism1. On the next day, NeutrAvidin beads were washed five times with pre-cooled RIPA buffer followed by proteins elution by boiling the beads in 2x loading buffer for 10 min. Proteins were subject to Western blotting and Commassie Blue staining followed by mass spectrometry for identification.

**In-gel trypsin digestion, LC-MS/MS, data processing and analysis.** Gel lanes were cut into slices and subjected to trypsin digestion. Briefly, gel slices were subjected to reduction and alkylation by TCEP (10 mM) and 2-chloroacetamide (55 mM). Protein was digested by incubating with trypsin (1 ng/µL) at 37 °C overnight, followed by extracting the tryptic peptides from the gel with 50% CAN / 5%FA, then 100% CAN sequentially. Peptide extracts were pooled, speedvac dried, followed by desalting using C18 StageTips for LC-MS/MS analysis.

Eluted peptides were analyzed with nanoelute UHPLC coupled to a Bruker TimsTOF protein mass spectrometer. Chromatographic separation was carried out using a linear gradient of 2–30% buffer B (0.1% FA in CAN) at a flow rate of 300 nl/min for 27 min. MS data was collected over a m/z range of 100 to 1700. During MS/MS data collection, each TIMS cycle was 1.1 s and included 1 MS + an average of 10 PASEF MS/MS scans.

Raw data were processed using MaxQuant 1.6.14.0 and searched against Mouse UniProt FASTA database. The settings are as below: oxidized methionine (M), acetylation (Protein N-term) were chosen as dynamic modifications, carbamidomethyl (C) as fixed modifications with minimum peptide length of 7 amino acids. Confident proteins were identified using a target-decoy approach with reversed database, strict false-discovery rate 1% at peptide and peptide spectrum matches (PSMs) level; minimum ≥1 unique peptide, ≥2 PSMs.

**Whole-mount immunostaining**
Embryonic kidneys were dissected and fixed overnight in 4% paraformaldehyde in 0.1% Triton X-100/PBS at 4 °C with gentle rotation. On the next day, samples were washed 3 times with PBTr, each time for 15 min, followed by blocking with 10% house serum and 3% BSA in 0.5% Triton-X100 at RT for 1–2 h with gentle rotation. Samples were then subjected to overnight incubation at 4 °C with primary antibodies prepared in 1:1 diluted blocking buffer. Samples were immunostained

using the following antibodies: CALB1 (1:400, C9848, Sigma), GFP (1:200, ab6556, Abcam), Pax2 (1:200, ab79389, Abcam), Cdh1 (1:400, #610181, BD), Alexa Fluor 568 donkey anti mouse IgG (1:500, Invitrogen, A10037) and Alexa Fluor 488 donkey anti rabbit IgG (1:500, Invitrogen, A32790). Before incubating with the secondary antibody (1:500) at RT for 1–2 h, samples were washed at least 3 times in PBTr. Samples were fine-dissected and oriented optimally before being mounted on slides with several drops of mounting medium. Images were acquired under Olympus BX-53 microscope.

**Histology and immunofluorescence staining**
Mouse embryonic kidney rudiments were fixed overnight in 4% PFA at 4 °C, embedded in gelatin for cryosection or in wax for paraffin section. Cryo-samples and paraffin samples were sectioned at 8 µm and 10 µm intervals, respectively. The sections were

blocked and penetrated for 1 h at room temperature in 3% FBS, 10% blood serum, and 0.2% triton-X100 followed by primary antibodies incubation overnight at 4 °C. Sections were immunostained using the following antibodies: CALB1 (1:400, C9848, Sigma), SIX2 (1:200, 11562-1-AP, Proteintech), GFP (1:200, ab6556, Abcam), Integrin α8 (1:400, AF4076, R&D), BrdU (1:100, #555627, BD), ETV5 (1:200, 13011-1-AP, Proteintech), phosphorylated-ERK (1:100, #4370, CST), cleaved-Caspase3 (1:200, #9661, CST), EphrinB1 (1:400, AF473, R&D), pFAK (1:300, #611722, BD), act Integrin β1 (1:100, #553715, BD) and N-cad (1:300, #610920, BD), Alexa Fluor 568 donkey anti mouse IgG (1:500, Invitrogen, A10037), Alexa Fluor 488 donkey anti rabbit IgG (1:500, Invitrogen, A32790), Alexa Fluor 594 donkey anti goat IgG (1:500, Invitrogen, A32758), Alexa Fluor 647 donkey anti rabbit IgG (1:500, Invitrogen, A31573) and Alexa Fluor 594 donkey anti rat IgG (1:500, Invitrogen, A21209). Sections were mounted with anti-fade mountant with DAPI (Invitrogene, S36938) and imaged under an Olympus BX-53 microscope or Zeiss LSM 800 confocal microscope.

**Whole-mount in situ hybridization**
RNA probes for in situ hybridization were prepared as previously described[63,64] Briefly, the probe plasmids were linearized, purified and used as template for in vitro transcription containing DIG labelling mix (11277073910, Roche). The template DNA was removed by DNase I and the DIG-labelled RNA probes were precipitated and eluted in RNase-free water.

Whole-mount in situ hybridization of embryos were performed as previously reported[65]. In brief, the fixed embryos were rehydrated in gradient methanol, treated with proteinase K, and post-fixed. The embryos were then pre-hybridized for 3 h before being incubated with a DIG-labelled probe overnight at 65 °C. Embryos were washed extensively followed by blocking in 10% BBR and 20% house serum for 3 h at RT. The the embryos were then incubated with anti-DIG-AP antibody (1:2000, 11093274910, Roche) at 4 °C overnight. After extensive washing, the AP activity would be detected by BM Purple (11442074001, Roche) for the desired time until the signal was developed optimally. After washing and fixing, the embryos were photographed on a Leica stereomicroscope MZ10F.

**BrdU incorporation assay**
Pregnant mice were intraperitoneally injected with BrdU solution at 1 mg per 10 g body weight, 2 h before embryo collection. BrdU solution was dissolved in PBS containing 15% DMSO. Embryonic tissues were collected and cryo-sections were prepared as described above. Before performing the immunostaining, slides were exposed to formalise-SSC solution followed by 2 N HCl acid treatment to denature the BrdU-incorporated DNA before blocking. The following procedures were the same as described in immunofluoresence staining. Sections were immunostained using BrdU antibody (1:100, #555627, BD) and Alexa Fluor 568 donkey anti mouse IgG (1:500, Invitrogen, A10037).

### In situ proximity ligation assay

Cryosections of embryonic kidney rudiments were processed for proximity ligation assay as previously described[66]. For the PLA labelling, probe anti-goat PLUS (Sigma, Duolink DUO92003), probe anti-rabbit MINUS (Sigma, Duolink DUO92004), and Duolink in situ detection reagent red (Sigma, Duolink DUO92008) were used. The following primary antibodies were used: rabbit anti-Ret antibody (1:200, Abcam, ab134100) goat anti-GDNF (1:400, R&D, AF212), rabbit anti-Ism1 antibody (1:300, Genescript) and goat anti-Integrin α8 (1:400, AF4076, R&D). Sections were mounted in the mounting medium with DAPI and subject to photograph using Zeiss LSM 800 confocal microscope.

### Migration assay

CMUB-1 cells ($5 \times 10^4$) were resuspended in 200 μL medium (1% FBS) and seeded in the upper well of the trans-well devices (6.5 mm diameter, 8 μm pore size, Corning Costar). WT or $Ism1^{-/-}$ CMMM-1 cells ($2 \times 10^5$) were resuspended in 600 μL serum-free medium and seeded in the lower well, in the presence or absence of rIsm1 (250 ng/ml). Cells were cultured for 48 h at 37 °C. After removing the non-migrated cells inside the membrane with a cotton swab, the migrated CMUB-1 cells on the outside of the trans-well membrane were fixed with 4% PFA and stained with 0.09% crystal violet solution. Image of migrated CMUB-1 cells were obtained under a microscope for five to eight random views. Cell numbers were counted using ImageJ.

### Hanging drop assay

CMMM-1 cells were washed with PBS and detached with 0.05% TE. Single cell suspensions were prepared at a concentration of $2.5 \times 10^6$ cells /ml in the presence of rIsm1 or other reagents. Deposit 10 μL drops onto the bottom of the lid of a 6 cm plate and the lid shall be turned over to allow the formation of cell aggregates and finally spheroids inside the drop. The spheroids formation was observed at different time points. For each group, at least 30 speroids were counted in an experiment. Three independent experiments were performed for analysis.

### Cell aggregation assay

Cell aggregation assays were performed as previously described[67]. Single cell suspensions were prepared in culture medium at a concentration of $10^5$ cells/ml in 1.5 ml Eppendorf tubes. For our experiment, function blocking antibody against β1 integrin (AIIB2) or the RGD peptide were added into the cell suspension, respectively. After pretreatment with AIIB2 or RGD for 30–45 min at 37 °C with 500 x g agitation, cells were subject to rIsm1 or Npnt and incubated for another 30–45 min with agitation. Cells were then seeded onto 24-well plates for 60 min at 37 °C in the incubator. Non-adherent cells were removed and the remaining cells were fixed and stained with crystal violet. Cell aggregation was measured using an inverted microscope by counting the number of aggregates (at least 5 cells makes an aggregates) in each field of a total three wells. The values of aggregates in each group were analyzed in comparison with the control group.

### Sample preparation for scRNA-seq

E10.5 or E11.5 kidney rudiments dissected from wild-type and $Ism1^{-/-}$ mice were digested in collagenase IV (1 mg/ml) at 37 °C for 20–30 min while shaking at 800 x g until no pellet was visible. Pipetting up and down every 5 min to make the digestion more thoroughly. The digested cells were then filtrated through a 40 μm strainer followed by a 800–1000 x g centrifugation for 3 min. Cells were washed with DPBS twice to remove fragments and finally resuspended in 50–100 μL of PBS with 0.04% BSA.

### Single-cell library construction and sequencing

Single-cell droplet libraries from the suspension were generated by the 10X Genomics Chromium Single Cell 5′ Library Construction Kit according to the manufacturer's instructions. The final libraries were sequenced by Annuogene on Illumina NovaSeq platform in 150 bp pair-ended manner.

### Processing raw sequencing data and cell-type clustering analysis

Raw sequencing reads were processed using the CellRanger software package (version 2.1.0) with default parameters from 10x Genomics. Data from multiple samples was aggregated and normalized to the same sequencing depth, resulting in a combined gene-barcode matrix among all samples. Further filtering and clustering analysis were performed with the Seurat R package.

### Statistics and reproducibility

For statistical analysis, figures are represented as the mean ± SD as indicated in figure legends. Data were analyzed using a two-tailed, unpaired Student's $t$ test by PRISM 9 Software. The criterion for statistical significance was $p < 0.05$ (*$p < 0.05$, **$p < 0.01$, ***$p < 0.001$ and ****$p < 0.00001$). Exact $p$ values were provided in figures or figure legends. Unless otherwise stated, the experiments were performed at least three times independently with similar results.

For quantification of cleaved-Caspase3 expressing or BrdU[+] cells, single-positive cells were counted in sections. Three to five random views were counted for each sample and 3–6 samples were stained in each group.

For the evaluation of branching morphogenesis in explants culture of kidney rudiments, the branching tips or branching points were quantified for each sample, and 3 samples in each group per experiment. The results were calculated from at least 3 independent experiments.

For the bright field images, immunostaining, in-situ hybridization images and gel/blot data, they were performed in at least three independent experiments with similar results, and all representative images reflect a minimum of three biological replicates.

### Reporting summary

Further information on research design is available in the Nature Portfolio Reporting Summary linked to this article.

## Data availability

The raw and processed data generated in this study have been deposited in the database under accession code PRJNA851535 (sc-RNA sequencing data) and PXD040642 (Mass Spectrometry data). Source data are provided with this paper.

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

## Acknowledgements
We thank Daniel Mak for proof reading of the entire manuscript, Lin-sheng Wang and Kaiqiang Zhao for critical discussions, and Sheung Kin Ken Wong, Huiling Zheng, WY Wong for technical assistance. This work was supported by grants from Science, Technology and Innovation Commission of Shenzhen Municipality (JCYJ20210324 114408024, Z.Z), the National Key Research and Development Program of China, Stem cell and Translational Research (2019YFA0111500, X.L); InnoHK@Health (Z.Z), Theme-based Research Scheme (Tl3-602/21-N, Z.Z), Guangdong High-level Hospital Construction Project (KJ012019517, Z.Z) and Guangdong Provincial People's Hospital Foundation (KY012021405), Guangdong-Dongguan Joint Research Scheme Guangdong-Hong Kong-Macau Program (2021B1515130004, Z.Z and G.J).

## Author contributions
Z.Z. conceived the project. G.G. and Z.Z. designed the experiments. G.G. performed the experiments. X.L. and G.G. analyzed the data. O.L., Z.J. and Y.T. initiated the investigation and helped to collect some of the data. X.Y. and G.J. contributed to discussion. All authors contributed to data interpretation and discussion. G.G. and Z.Z. prepared the manuscript with inputs from all authors.

## Competing interests
The authors declare no competing interests.
