## [Peer Review File · Nature Communications]

Isthmin-1 (Ism1) modulates renal branching morphogenesis and mesenchyme condensation during early kidney developmentREVIEWER COMMENTS

Reviewer #1 (Remarks to the Author):

Review: Isthmin-1 is required for renal branching morphogenesis by promoting Gdnf/Ret signaling and mesenchyme condensation during early kidney development
Gao et al. – Nature Communications

In this manuscript, the authors use a variety of in vivo and invitro experimental approaches to argue that the secreted factor Ism1 is a ligand for integrin receptors mediating condensation and signaling by the nephrogenic mesenchyme at the outset of mammalian kidney development. When this interaction is lost, by conditional removal of Ism1 encoding sequence, there is a variable, early kidney agenesis phenotype, consistent with Ism1 regulation of early kidney development. Overall, there are a wide-range of approaches adopted by the authors. However, not all of these are really informative, and perhaps by trying to cover too much, key procedures and caveats to interpretations are glossed over. The work is interesting and the in vitro experiments in particular add interesting mechanistic insight.

The paper starts with the authors efforts in Figure 1 and 2 to characterize early kidney development through scRNA-seq. This is perhaps the weakest part of the paper. The evidence for the annotation of certain clusters in d is scant and the schematic in 1b is almost certainly incorrect. If the authors really think this data adds to the paper, then there should be rigorous mapping back of key markers to kidney anlagen. In the text, how does Shh allantoic expression relate to the UE2 population. Isn't this simply known Shh expression in the stalk region of the UE. If a population of cells is thought to be neural crest, why label these spinal cord? Stroma and intercalated cells are often used interchangeably, but here IC cells are singled out as population I don't think exists in the position shown in Figure 1B. Similarly, there is no hard evidence to connect the zones indicated in Figure 1i to clusters in Figure 1h and the distal and transitional clusters may simply represent proliferating or poor quality cell types. In Figure 2, the authors point out that only GDNF and RET signaling comes out as clear ligand receptor interaction in the informatics of mesenchyme and epithelium. But, knowing a role and relevant expression for Fgf10 and FGFR, does this indicate a weakness in this approach. In short, there is a not very compelling scRNA-seq study that in the end focuses on Ism1. The data collected by the authors can be useful to the community, but perhaps to the extent the authors need to use this data, Figures 1 and 2 could be reduced to a single figure and the authors can avoid over speculation and stick to the purpose of extracting Ism1 from these data and documenting its expression. With respect to Ism1 activity revealed by the GFP for the targeted allele and for what the authors claim for in situ, there are quite significant differences. The authors should address these and reasonable explanations. As an example, there is clearly strong GFP reporting on activity of the targeted Ism1 allele in both the nephrogenic mesenchyme and invading ureteric branch tip at E12.5 though the authors state that Ism1 expression is lost from the later 24 hours earlier. Clearly, the secreted nature of Ism1 and the co-expression in the two interacting cell populations driving early kidney development leaves open the door for several possibilities as to the target cell type..

Figure 3 presents the statistics for kidney agenesis. A puzzle that goes unmentioned is half the collected embryos are not expected to exhibit a phenotype given the absence of an adult phenotype and one might imagine this as a complicating factor (as it so often is with other non-fully penetrant agenesis mutants) in interpreting events. What is delayed normal development (all litters have a range of stages that may vary in developmental terms by 12 hours) and what is the actual developmental phenotype. That this concern is never addressed is concerning? In the data in vitro, Figure 3m and 3n, it also looks as if the phenotype is far more penetrant in culture? Is this the case? And, why? Note – the authors assertion on line 294 that "Ism1 is indispensable for kidney branching morphogenesis" is clearly not correct when more than half the kidneys develop fine (apparently) with no Ism1.

The clear experiment here is adding recombinant Ism1 to the medium of kidney explants rescues the Ism1 mutant phenotype – a strong result. In contrast, the experiments in Figure 4i-m are not. Is the dissociated epithelial cell migration a reasonable assay to investigate collective migration of the ureteric epithelium. This seems a stretch. Is the effect of rIsm1 on the viability of cells which is indirectly measured by enhanced motility?

Figure 5 uses scRNA-seq to examine gene activity in Ism1 mutant cells. The better way of displaying this data would be to show the relative levels of gene activity within each of the specific sub clusters. It is not clear what the data in Figure 5b represents?

All-in-all looking at the insitu and immuno expression data in several figures one is left with the general view that all phenotypes can be explained by a failure of normal ingrowth of the ureteric epithelium which then results in the failure to modify gene activity within the UE epithelium itself into clear tip and stalk domains and a failure of condensation and capping of the metanephric mesenchyme requiring a physical contact with the epithelium. Here, the most informative experiment is the Etv5 and phospho-Erk analysis in Figure 6. This makes very clear that one would not expect the ureteric epithelium to grow. The authors can strength this data by addressing whether Ism1, Gdnf and Fgf10 can rescue Etv5 and phosphor-Erk in culture. One expects so. This would support the authors contention that it is “weakened” branch-growth signaling that underlies the primary phenotype.

The Western data in particular in Figures 7 and 8 provide robust evidence to support the model that rIsm1 signals through the integrin $\alpha 8$ receptor. Supporting a model that rIsm1 signals directly to the nephrogenic mesenchyme, promoting mesenchymal programs that secondarily effects the epitheliums ingrowth – such as production of GDNF. Unfortunately, experiments to more directly examine this with isolated nephrogenic mesenchyme from mutants at e11.5 may be quite challenging though perhaps not so using mesenchyme at a later stage though responses could differ at a later time. Can the specific phosphorylation of FAK and downstream be observed in immuno on sections. Could this proceed loss of phosphor-ERK/Etv5 in the branch tip as providing more evidence for the primary effect on signaling within the nephrogenic mesenchyme? Exploring Ism1 in isolated UE culture could clarify further. One would expect no independent and no synergistic action with GDNF absent the nephrogenic mesenchyme.

In summary, this study can be improved to reduce non-informative information, to clarify discrepancies in the phenotype in vivo and in vitro and balance conclusions, and if additional insight is possible, to use other approaches to support the nephrogenic mesenchyme/Itga8 primary signaling role in the normal auto-regulation of Gdnf levels.

Minor Comments

- 1) On line 72, there appears to be a typo “Renal genesis” should read “Renal agenesis.”
- 2) On line 91, “MHB” should be “midbrain-hindbrain boundary” to be accessible to a broader audience, rather than waiting until line 260 to define the abbreviation.
- 3) On line 97, the text “Ism1, initially identified...secreted protein” is repeating the first sentence of the paragraph and should be changed.
- 4) In figure 2c, the first panel says e10.75, but the text on line 247 referencing this figure says “e10.5”
- 5) On line 256, the authors describe the generation of the Ism conditional knockout mice, but only describe the insertion of the first loxP site.
- 6) On line 302, the authors state that the “epithelium did not enter the mesenchyme;” it would help to have an outline of where the mesenchyme lies in the corresponding figures without nuclear co-staining. It would also be helpful to have “Calb1” written in the figure itself.
- 7) In figures 3h-k, it may be better to harmonize the text and the figure headings by consistently referring to the recombinant Ism protein as “rIsm” in the figure or “recombinant Ism protein” in the

text.

8) On line 367, the authors should define what CMUB-1 and CMMM-1 cell lines are.

9) In figure 5a, the authors conclude that because the number of cells in NPC clusters are reduced in *Ism* mutants, there is a loss of nephron progenitor cells, however, there is no quantification or bar plot showing a reduction in the proportion of cells from the null mice.

10) In figure 5f, outlining the ureteric bud/tips would be helpful on wild type samples.

11) In figure 5h, the bar graphs should be labeled using genotypes to be consistent with the rest of the paper. Labels as is can be misconstrued.

12) In figure 6g, "NC" should be defined with consistent labeling to other figures.

13) In line 518, the authors state that "

.

Reviewer #2 (Remarks to the Author):

This is an interesting paper that suggests a novel mechanism involved in ureteric bud branching during kidney development. The authors then go on to propose a mechanism involving $\alpha 8\beta 1$ integrin, ret signaling and mesenchymal morphogenesis.

Main criticism:

The mechanism, while interesting and potentially correct, does not seem fully supported yet.

There are well established systems in rodents for studying in vitro isolated ureteric bud branching in 3D matrix, isolated Wolffian duct budding and isolated metanephric mesenchyme morphogenesis. These systems were particularly useful for clarifying the roles of growth factors and integrins in various aspects of early kidney development in relative isolation from complexities of mutual inductive events. Nevertheless, these systems can be made to undergo a sufficient amount of morphogenesis and differentiation in isolation, and further development occurs in recombination experiments with mesenchyme.

The mechanism that the authors propose would seem to make one or more predictions with these 3 isolated systems (with mesenchyme recombination) and therefore should be analyzed accordingly in WT and KO.

By following appropriate markers in WT vs KO, such studies are likely to either support the proposed mechanism or result in a revision of the mechanism. If such a revision is well supported, that may be fine.

Reviewer #3 (Remarks to the Author):

This is a great paper that is derived from a preliminary single cell analysis. To identify a new factor that functions through the integrin pathway to regulate kidney morphogenesis through cell-cell adhesion is exemplary. One distraction from the story is the pseudotime trajectory analysis of the single cell data. In most cases it is unsupported by what the community knows of the biology of nephrogenesis and branching morphogenesis. This should not take away the importance of the findings, however, which are significant.

The description of the initial ureteric budding and first bifurcation is confusing. The nephric duct, after budding, is dispensable for kidney development as evidenced by many transwell culture experiments over decades - where the bud is separated from the duct (but contained within the metanephric

mesenchyme) it continues to grow and proliferate unhindered. It is this reviewer's opinion that the nephric duct should be removed from the analysis. Furthermore, the use of Monocle 2 to establish a developmental trajectory is problematic. While trajectory analysis can be a very useful tool to dissect the continuum of cellular development and differentiation, Monocle 2 typically favours a forced bifurcated trajectory when perhaps none exists (the default settings are set for a single branchpoint). I think this holds true for the UE1 analysis in the first result. The ureteric bud sprouts from the nephric duct, from there the bud drives growth of the ureteric collecting duct without contribution from the nephric duct. The "ureteric tree" or collecting duct network is the result of migration of UB tip cells through to trunk and finally the branches as growth continues (later, in situ proliferation is a driving force for tubular extension as well). The authors, using Monocle 2 (now deprecated & outdated), describe a trajectory beginning with nephric duct, which forks into a ureteric bud fate, and a distinct ureteric trunk fate. However, Nephric duct first forms ureteric bud, and the bud differentiates into ureteric trunk (future collecting duct). This differentiation mechanism is biologically established as a linear trajectory using UB cell tracking and multiple fate analysis experiments.

There needs to be a better justification for the clustering performed in the "Heterogeneity of nephrogenic progenitors" section (line 171). Subclustering the NPC cluster into eight clusters is really pushing too far given the the markers identified to justify each cluster (Ext Fig 3b). Now that we have alternative modalities to do site-specific gene expression analysis using 10X spatial transcriptomics (for example), this manuscript should use that approach in order to justify this degree of clustering. Can the authors include classical NPC markers in the plot in Ext Fig 3b? One of the clusters seems to be highly proliferative, as evidenced by the top expressed genes in Ext Fig 3b, and this might agree with Ext Fig 1b. A DimPlot with the cell cycle associations as labels is required for the subclustered NPC pool (Fig1h maybe?), this will help delineate what the different NPC clusters are doing. The "distal" NPC pool might actually be classical "metanephric mesenchyme" or semi-derived intermediate mesoderm (which is actually shown in Figure 3e) – can some markers for those cell types also be provided in Ext Fig 3b?

The pseudotime trajectory analysis with Monocle 2 is troublesome with the NPCs as well. In Fig 1J, the "distal" or "transition" cells differentiate into two main pools – the "differentiated" and "undifferentiated". But this lacks biological evidence. What we know about NPC commitment, is that a slow cycling NPC pool typically where in Fig 1i the "CM/OriNPC" cells are. These cells then become committed (where the indNPCs are) before transitioning below the tip and forming pre-tubular aggregates soon after (where the diffNPC cells are). The "distal" and "transition" cells are perhaps metanephric mesenchyme or intermediate mesoderm at this E10.5/E11.5 age, but the trajectory analysis having them directly contribute in almost equal proportions to both committed and non-committed NPC fates (Fig1j) is hard to believe.

In the "Analysis of the cross talk" section, on line 227 the authors state "GDNF stood out as the only ligand produced from NPCs and received by UTip cells (Fig 2a)". This is patently untrue, the figure shows a number of other ligands being upregulated in both (IL2, FGF, PTN). Regardless, the ongoing investigation using GDNF is still justified and yields interesting results. The authors could simply reword "stood out" to "was the most significant" (as an example).

The section on the Ism1 conditional knockout is great, except for a statement where the authors state on line 270/271/272 "Based on the data from 170 Ism1^{-/-} embryos collected between E14.5 and P0, approximately 60% of homozygous mutants exhibited various kidney defects". Followed by, on line 274, "These results demonstrated that Ism1 is essential for kidney development". How is that so, when 40% of the homozygous embryos collected between E14.5 and P0 had no phenotype? How can a kidney form normally in 40% of the embryos/pups that have conditionally removed Ism1? Is alternative splicing an issues such that the KO region is spliced out? Or is there an issue with recombination efficiency? Or functional redundancy? I feel that this must be addressed & explained if the authors are going to state that it is "essential" for kidney development. The authors bring this up in the discussion briefly, but the statements in the results need to be tempered down. The authors also say on line 546 in the Discussion "the penetration was even higher (more than 80%) when examined at E11.5". But that is not in the results section. I don't understand how the authors' Ism1 mice can have 80% penetrance at E11.5, and only 60% penetrance between E14.5 and P0. In this same result, the authors explain that Ism1 deficient kidney rudiments fail to enter the

metanephric mesenchyme and no "t" stage is reached. This is discussed in a way that it makes the reader think that this failure is a standard feature of Ism1 loss. However, in the phenotype we are shown that - while there is indeed agenesis - there is also kidney hypoplasia where that first budding stage has occurred, but not been particularly successful. Further, there is *unilateral* agenesis - so if an absent kidney has a contralateral kidney that is "normal", why isn't Ism1 loss having the same effect there? This comes back to the matter of whether Ism1 can be considered "essential" for kidney development. The phenotype is no doubt striking, very impressive indeed, but I think it must be described in different terms.

Figure 4b - the WT dots on the plot are obfuscating the Ism1^{-/-} dots; so it is hard to compare the relative distribution of the two.

Extended Data Fig6b, 6d, 6e. Should split the plots by genotype.. i.e. [split.by = "Genotype"] in Seurat. Then the Cell Cycle, Ube2c can be compared between the WT & Mut phenotypes, and the Dimplot in Ext 6b can be better compared between the two genotypes (the pink and blue dots overlaid can erroneously imply an abundance of one cell type from one genotype over another). Splitting by genotype will fix that impression.

Line 330 - what do the authors mean by "Concatenating the datasets from two genotypes at two time points"?

Line 337 onwards is poorly explained.

Line 340 - what do the authors mean by "new trunk" and "new tip" cells? How are the authors classifying these cells? What is the difference between "tip" and "bud"; they both are Ret⁺, Gfra⁺. The authors haven't shown the subclusters, and in the subclustering, how are the authors delineating "BudTip", "PreBud", "PreTip", "Trunk" and "New Trunk", in the Seurat object the authors call them "cluster_name"? What representative genes are the authors relying on to classify these cells as belonging to these structures? I think this work needs to be redone using spatial transcriptomics, only then can the authors classify "Prebud" from "BudTip" (Fig 4c). What does "Pre bud" even mean? If the authors mean the first bud from the nephric duct, then that isn't present in the E11.5 kidney. Figure 4 only represents the E11.5 data. Like before, the Fig4B plot needs to be split by genotype - overlaying the blue and pink dots is insufficient if the reader is required to use it as evidence for a difference in distribution across the pseudotime calculated by Monocle 2.0.

On the UE1 pseudotime in Figure 4, Monocle 2.0 seems to be doing something odd. Transcriptionally, "New Tip" (Fig4d) would have almost the same developmental plasticity and identity as the "bud" (4d) - however, it has placed the "new tip" population at the end of pseudotime. Why? The authors seem to agree with what I say here when they state on line 343 - "the identity of new tips cells closely resembled that of bud cells". Of course, they do. Monocle is again used in Fig5a, and the previous comments in this review which question the trajectory calculated by Monocle 2.0 for the NPCs (Fig1j) still stand in this instance. In Figure 5a, the trajectory work is probably dispensable for the manuscript as it doesn't add any significance to the work (in the way it is currently presented, at least).

A better description of how the bioinformatic integration was done to combine and then analyse E10.5 and E11.5 datasets is required. Were all the cells just processed as one dataset, or was each age independently classified, and then a label transfer-based integration performed?

In my opinion, the authors need to revisit the pseudotime analysis, perhaps using an alternative package such as Sandrine Dudoit's Slingshot to support the trajectories generated by Monocle 2. At the moment, the trajectories in the manuscript do not represent the biology known about the differentiation processes being described.

While the trajectory analysis and potential overclustering is problematic, the differential gene expression analysis is fine and the application of CellChat yields rich data which is put to good use biologically. From here on, the paper is great and insightful and enjoyable to read. Experiments are well justified, and the conclusions well supported.

Point-to-Point response to the reviewers

Response to Reviewer #1 (Remarks to the Author):

Review: Isthmin-1 is required for renal branching morphogenesis by promoting Gdnf/Ret signaling and mesenchyme condensation during early kidney development
Gao et al. – Nature Communications

In this manuscript, the authors use a variety of in vivo and invitro experimental approaches to argue that the secreted factor *Ism1* is a ligand for integrin receptors mediating condensation and signaling by the nephrogenic mesenchyme at the outset of mammalian kidney development. When this interaction is lost, by conditional removal of *Ism1* encoding sequence, there is a variable, early kidney agenesis phenotype, consistent with *Ism1* regulation of early kidney development. Overall, there are a wide-range of approaches adopted by the authors. However, not all of these are really informative, and perhaps by trying to cover too much, key procedures and caveats to interpretations are glossed over. The work is interesting and the in vitro experiments in particular add interesting mechanistic insight.

The paper starts with the authors efforts in Figure 1 and 2 to characterize early kidney development through scRNA-seq. This is perhaps the weakest part of the paper. The evidence for the annotation of certain clusters in d is scant and the schematic in 1b is almost certainly incorrect. If the authors really think this data adds to the paper, then there should be rigorous mapping back of key markers to kidney anlagen. In the text, how does *Shh* allantoic expression relate to the UE2 population. Isn't this simply known *Shh* expression in the stalk region of the UE. If a population of cells is thought to be neural crest, why label these spinal cord? Stroma and intercalated cells are often used interchangeably, but here IC cells are singled out as population I don't think exists in the position shown in Figure 1B. Similarly, ***there is no hard evidence to connect the zones indicated in Figure 1i to clusters in Figure 1h and the distal and transitional clusters may simply represent proliferating or poor quality cell types.***

In Figure 2, the authors point out that only GDNF and RET signaling comes out as clear ligand receptor interaction in the informatics of mesenchyme and epithelium. But, ***knowing a role and relevant expression for *Fgf10* and *FGFR*, does this indicate a weakness in this approach.*** In short, there is a not very compelling scRNA-seq study that in the end focuses on *Ism1*. The data collected by the authors can be useful to the community, but perhaps to the extent the authors need to use this data, ***Figures 1 and 2 could be reduced***

to a single figure and the authors can avoid over speculation and stick to the purpose of extracting *Ism1* from these data and documenting its expression.

With respect to *Ism1* activity revealed by the GFP for the targeted allele and for what the authors claim for *in situ*, there are quite significant differences. The authors should address these and reasonable explanations. As an example, there is clearly strong GFP reporting on activity of the targeted *Ism1* allele in both the nephrogenic mesenchyme and invading ureteric branch tip at E12.5 though the authors state that *Ism1* expression is lost from the later 24 hours earlier. Clearly, the secreted nature of *Ism1* and the co-expression in the two interacting cell populations driving early kidney development leaves open the door for several possibilities as to the target cell type..

Response:

We thank the reviewer for the constructive and insightful comments. Following the reviewer's advice, we have combined the Figure 1 and Figure 2 into one figure (Figure 1) in the revised manuscript and emphasized the identification of *Ism1* as an important ligand involving in the communication between mesenchyme and epithelium.

Shh is expressed in the cloacal epithelium, internal urethra, bladder and urethral plate in developing urogenital tissues (Haraguchi, R., J. Motoyama, H. Sasaki, Y. Satoh, S. Miyagawa, N. Nakagata, A. Moon and G. Yamada (2007). "Molecular analysis of coordinated bladder and urogenital organ formation by Hedgehog signaling." *Development* **134**(3): 525-533.) and marks the potential progenitor cells in adult ureter (Fink, E. E., S. Sona, U. Tran, P. E. Desprez, M. Bradley, H. Qiu, M. Eltemamy, A. Wee, M. Wolkov, M. Nicolas, B. Min, G. P. Haber, O. Wessely, B. H. Lee and A. H. Ting (2022). "Single-cell and spatial mapping Identify cell types and signaling Networks in the human ureter." *Dev Cell* **57**(15): 1899-1916 e1896.). We therefore changed the label from UE2 to Ureter epithelial Cells(UrECs) (Figure 1a).

We also changed the labelling of neural crest (NCs). We agree that the labelling of various cell population may not be accurate, given that the kidney is not yet an independent tissue at E10.5-E11.5 with mixed populations of mesenchyme, we deleted this figure in the revised manuscript.

Our analysis did show the existence of weak FGFs/FGFR signaling in the crosstalk between mesenchyme and epithelium. FGF ligands were produced from the majority of the NPC sub-clusters, especially CapM and MM, and received by both UE and NPC. However, the FGF/FGFR signaling mediates the crosstalk predominantly between the NPCs sub-clusters. Given the existence of complex FGF ligands, we further analyzed *Fgf10/Fgfr1/2* in these

subclusters as shown in the figure below.

At E10.75, *Ism1* was weakly detectable in the nephrogenic epithelium close to nephric duct but strongly expressed in mesenchyme by *in situ* hybridization. The staining of GFP in the epithelium by immunostaining at the same time may represent un-degraded GFP as it is common that GFP has a much longer half-life and is quite stable than mRNA. This is consistent with our RNA-seq data that *Ism1* was mainly found in mesenchyme (**Extended Figure 3b**).

Figure 3 presents the statistics for kidney agenesis. A puzzle that goes unmentioned is half the collected embryos are not expected to exhibit a phenotype given the absence of an adult phenotype and one might imagine this as a complicating factor (as it so often is with other non-fully penetrant agenesis mutants) in interpreting events. ***What is delayed normal development (all litters have a range of stages that may vary in developmental terms by 12 hours) and what is the actual developmental phenotype. That this concern is never addressed is concerning?*** In the data *in vitro*, **Figure 3m and 3n**, it also looks as if the phenotype is far more penetrant in culture? Is this the case? And, why? Note – the authors assertion on line 294 that “*Ism1* is indispensable for kidney branching morphogenesis” is clearly not correct when more than half the kidneys develop fine (apparently) with no *Ism1*.

Yes, delayed development did exist, even in the wild-type mice. As illustrated in **Extended Fig.12**, five different types of UB morphology could be observed in the mutant at E11.5. Type I represents normal UB branching which accounts for about 20%. Among the 80% UB without branching structure, 20% were type II UB which was also appeared in wild-type and heterozygous embryos though with significant lower percentages (11% and 15%, respectively). Type II represents those with delayed development and could further developed into kidney. These percentage coincides well with the observed percentage of defective branching morphogenesis at E11.5 and defective kidney development at birth.

The phenotype was more penetrant in explant culture because the majority kidney rudiments taken for *ex vivo* culture were type III-V.

For the assertion on line 294 that “*lsm1* is indispensable for kidney branching morphogenesis”, we have changed to “required” to tone down in the revised manuscript.

The clear experiment here is adding recombinant *lsm1* to the medium of kidney explants rescues the *lsm1* mutant phenotype – a strong result. ***In contrast, the experiments in Figure 4i-m are not. Is the dissociated epithelial cell migration a reasonable assay to investigate collective migration of the ureteric epithelium. This seems a stretch. Is the effect of *rlsm1* on the viability of cells which is indirectly measured by enhanced motility?***

We are pleased that the reviewer considered the explant culture is a strong result. We agree that epithelial cell migration may not fully represent the *in vivo* situation. We tried to explore one of the aspects that can affect epithelial invasion into mesenchyme. The effect of *rlsm1* on epithelial migration does not seem to be associated with cell viability as *ISM1* did not significantly affect the survival of mesenchyme cells or ureteric epithelial cells in MTT assay (see data below).

Figure 5 uses scRNA-seq to examine gene activity in *lsm1* mutant cells. ***The better way of displaying this data would be to show the relative levels of gene activity within each of the specific sub clusters. It is not clear what the data in Figure 5b represents?***

We now show some of the important DEGs between different genotypes, as suggested in the revised manuscript (Figure 4c).

All-in-all looking at the *in situ* and immuno expression data in several figures one is left with the general view that all phenotypes can be explained by a failure of normal ingrowth of the ureteric epithelium which then results in the failure to modify gene activity within the UE epithelium itself into clear tip and stalk domains and a failure of condensation and capping of

the metanephric mesenchyme requiring a physical contact with the epithelium. Here, the most informative experiment is the Etv5 and phospho-Erk analysis in Figure 6. This makes very clear that one would not expect the ureteric epithelium to grow. **The authors can strength this data by addressing whether *Ism1*, *Gdnf* and *Fgf10* can rescue *Etv5* and phospho-Erk in culture. One expects so. This would support the authors contention that it is “weakened” branch-growth signaling that underlies the primary phenotype.** We are pleased that the reviewer consider the Figure 6 is informative. Following the reviewer’s suggestion, we now include Etv5 and p-Erk data in response to *Ism1*, *Gdnf*, and *Fgf10* in the *ex vivo* culture (Figure 5h). As expected, all these ligands can rescue the Etv5 and p-Erk expression.

The Western data in particular in Figures 7 and 8 provide robust evidence to support the model that *rIsm1* signals through the integrin $\alpha 8$ receptor. Supporting a model that *rIsm1* signals directly to the nephrogenic mesenchyme, promoting mesenchymal programs that secondarily effects the epithelium ingrowth – such as production of GDNF. **Unfortunately, experiments to more directly examine this with isolated nephrogenic mesenchyme from mutants at e11.5 may be quite challenging though perhaps not so using mesenchyme at a later stage though responses could differ at a later time.**

We are very pleased that the reviewer considered Figure 7 and 8 provide robust evidence to

support our model. Following the reviewer's advice, we isolated the nephrogenic mesenchyme from mutants at E11.5 and cultured in the presence or absence of *rism1* to detect the expression of *Gdnf* and *Six2*. Below is the bright field image of isolated E11.5 *Ism1*^{-/-} mesenchymal cells cultured as monolayer in plate. Cells were plated for 24 hours followed by 48 hours culture in the presence or absence of *Ism1* before harvesting for RNA extraction. The expression of *Gdnf* and *Six2* were examined by qPCR. As shown in Figure below, *Gdnf* and *Six2* were both significantly up-regulated. These results are now included in the revised manuscript (**Figure 5g**).

Can the specific phosphorylation of FAK and downstream be observed in immuno on sections. Could this proceed loss of phosphor-ERK/Etv5 in the branch tip as providing more evidence for the primary effect on signaling within the nephrogenic mesenchyme? Exploring Ism1 in isolated UE culture could clarify further. One would expect no independent and no synergistic action with GDNF absent the nephrogenic mesenchyme. We examined the phosphorylation of FAK on sections. As shown in Figure below, pFAK is lost in the mesenchyme adjacent to the epithelium in *Ism1* deficient kidney rudiment at E11.5 embryos. This data is now included in the revised manuscript as **Figure 7h**. We also examined the Integrin signaling in E11.5 isolated mesenchyme cells, in the presence or absence of *rism1*. As shown in **Figure 7i** and **7j**, both *Cdc42* expression and the phosphorylation of FAK were upregulated in the presence of *rism1*, supporting its role in activating Integrin signaling.

To explore the possibility that *Ism1* may directly affect UE branching morphogenesis, we dissect UB from the surrounding mesenchyme and culture in Matrigel to see if exogenous *rism1* could promote its branching morphogenesis. As shown below, in the absence of mesenchyme, UB isolated from either wild-type or *Ism1* mutant embryos exhibited similar branching morphogenesis in the 3D culture system (a, b), in the presence or absence of *rism1* (c). As GDNF is required in this culture system (Zhang, X., K. T. Bush and S. K. Nigam (2012). "In vitro culture of embryonic kidney rudiments and isolated ureteric buds." *Methods Mol Biol* **886**: 13-21.), it is plausible that exogenous GDNF is sufficient to activates Ret signaling which is required for UB branching morphogenesis.

In summary, this study can be improved to reduce non-informative information, to clarify discrepancies in the phenotype in vivo and in vitro and balance conclusions, and if additional insight is possible, to use other approaches to support the nephrogenic mesenchyme/*Itga8* primary signaling role in the normal auto-regulation of *Gdnf* levels.

Thanks for the advice and we have combined the Figure 1 and 2 and reduced the non-informative data especially the single cell RNA-seq data to make the manuscript more focused.

Minor Comments

1) On line 72, there appears to be a typo “Renal genesis” should read “Renal agenesis.”

Thanks and we have revised in the manuscript.

2) On line 91, “MHB” should be “midbrain-hindbrain boundary” to be accessible to a broader audience, rather than waiting until line 260 to define the abbreviation.

We have revised in the manuscript.

3) On line 97, the text “*lsm1*, initially identified...secreted protein” is repeating the first sentence of the paragraph and should be changed.

We have deleted the sentence in the revised manuscript.

4) In figure 2c, the first panel says e10.75, but the text on line 247 referencing this figure says “e10.5”

We have changed it to E10.75 in the revised manuscript.

5) On line 256, the authors describe the generation of the *lsm* conditional knockout mice, but only describe the insertion of the first loxP site.

This is now described in the revised manuscript.

6) On line 302, the authors state that the “epithelium did not enter the mesenchyme;” it would help to have an outline of where the mesenchyme lies in the corresponding figures without nuclear co-staining. It would also be helpful to have “*Calb1*” written in the figure itself.

Both outline and “*Calb*” were added in the revised Figure 2f.

7) In figures 3h-k, it may be better to harmonize the text and the figure headings by

consistently referring to the recombinant Ism protein as “rIsm” in the figure or “ recombinant Ism protein” in the text.

We have changed in the revised Figure 2h-k.

8) On line 367, the authors should define what CMUB-1 and CMMM-1 cell lines are.

The description of CMUB-1 and CMMM-1 cell lines was involved in the revised manuscript.

9) In figure 5a, the authors conclude that because the number of cells in NPC clusters are reduced in Ism mutants, there is a loss of nephron progenitor cells, however, there is no quantification or bar plot showing a reduction in the proportion of cells from the null mice.

This part was added in the revised Figure (Extended Fig.8b) and the percentage of CapM was decreased in E10.5 *Ism*^{1-/-} sample compared with that in WT mice.

Supplementary

10) In figure 5f, outlining the ureteric bud/tips would be helpful on wild type samples.

The outline was added in the revised Fig.4f.

11) In figure 5h, the bar graphs should be labeled using genotypes to be consistent with the rest of the paper. Labels as is can be misconstrued.

The labels have been changed as suggested in the revised Fig.4h.

12) In figure 6g, “NC” should be defined with consistent labeling to other figures.

It has been changed in the revised Fig.6e.

13) In line 518, the authors state that “

Response to Reviewer #2 (Remarks to the Author):

This is an interesting paper that suggests a novel mechanism involved in ureteric bud branching during kidney development. The authors then go on to propose a mechanism involving $\alpha 8 \beta 1$ integrin, ret signaling and mesenchymal morphogenesis.

Main criticism:

The mechanism, while interesting and potentially correct, does not seem fully supported yet.

There are well established systems in rodents for studying in vitro isolated ureteric bud branching in 3D matrix, isolated Wolffian duct budding and isolated metanephric mesenchyme morphogenesis. These systems were particularly useful for clarifying the roles of growth factors and integrins in various aspects of early kidney development in relative isolation from complexities of mutual inductive events. Nevertheless, these systems can be made to undergo a sufficient amount of morphogenesis and differentiation in isolation, and further development occurs in recombination experiments with mesenchyme.

The mechanism that the authors propose would seem to make one or more predictions with these 3 isolated systems (with mesenchyme recombination) and therefore should be analyzed accordingly in WT and KO.

By following appropriate markers in WT vs KO, such studies are likely to either support the proposed mechanism or result in a revision of the mechanism. If such a revision is well supported, that may be fine.

Response to reviewer's comments:

Thanks for the encouraging and insightful comments. Following the advice, we have performed recombinant culture with isolated UB and MM to assess if *Ism1* facilitates kidney branching through regulating Integrin signaling. As ureteric bud formation is not affected in the *Ism1* mutant embryo, we focused on the effect of *ISM1* on UB branching and integrin signaling in mesenchyme.

Firstly, we confirmed the role of *Ism1* in promoting Integrin signaling in E11.5 mesenchyme. The mesenchyme cells isolated from *Ism1* deficient kidney rudiments exhibited impaired Integrin signaling, as shown in **Figure 7h** and **Extended Fig. 11b**. The downstream targets, phosphorylation of FAK and the expression of RhoA, Rac1, Cdc42 were all downregulated in E11.5 mutant mesenchyme. To further examine if exogenous *rlsm1* could activate or restore

Integrin signaling, the expression of *Cdc42*, *Gdnf* and phosphorylation of FAK were detected in E11.5^{-/-} mesenchyme dissected from UB in the presence or absence of rIsm1. As shown in Figure 5g, 7i and 7j, Integrin signaling in isolated E11.5^{-/-} mesenchyme was upregulated upon exogenous rIsm1, supporting the role of Ism1 in promoting Integrin signaling in E11.5 mesenchyme.

Next, we examined the possibility that Ism1 directly function on isolated UB and promote its branching morphogenesis. As shown in the Figure below, in the absence of mesenchyme, UB isolated from either wild-type or *Ism1* mutant embryos exhibited similar branching morphogenesis in the 3D culture system (a, b), in the presence or absence of rIsm1 (c). As GDNF is required in this culture system (Zhang, X., K. T. Bush and S. K. Nigam (2012). "In vitro culture of embryonic kidney rudiments and isolated ureteric buds." *Methods Mol Biol* 886: 13-21.), it is plausible that exogenous GDNF is sufficient to activates Ret signaling which is required for UB branching morphogenesis.

We also performed a recombinant assay using isolated UB and MM from WT and *Ism1*-null mice in a 3D culture system. As shown below (a), WT UB can branch in the Matrigel in recombinant with WT MM, but not with *Ism1*-null MM. Exogenous rIsm1 can promote WT UB branching in recombinant with either WT or *Ism1*-null MM. Similar results were observed in E10.75 *Ism1*-null UB culturing in Matrigel (b). WT MM can rescue the branching defect in recombinant with *Ism1*-null UB and *Ism1*-null MM would recapitulate the branching defect even with WT UB, indicating that MM with normal Ism1 expression is required for UB branching morphogenesis.

E10.75 WT UB

E10.75 *lsm1*^{-/-} UB

Response to Reviewer #3 (Remarks to the Author):

This is a great paper that is derived from a preliminary single cell analysis. To identify a new factor that functions through the integrin pathway to regulate kidney morphogenesis through cell-cell adhesion is exemplary. One distraction from the story is the pseudotime trajectory analysis of the single cell data. In most cases it is unsupported by what the community knows of the biology of nephrogenesis and branching morphogenesis. This should not take away the importance of the findings, however, which are significant.

The description of the initial ureteric budding and first bifurcation is confusing. The nephric duct, after budding, is dispensable for kidney development as evidenced by many transwell culture experiments over decades - where the bud is separated from the duct (but contained within the metanephric mesenchyme) it continues to grow and proliferate unhindered. ***It is this reviewer's opinion that the nephric duct should be removed from the analysis. Furthermore, the use of Monocle 2 to establish a developmental trajectory is problematic.*** While trajectory analysis can be a very useful tool to dissect the continuum of cellular development and differentiation, Monocle 2 typically favours a forced bifurcated trajectory when perhaps none exists (the default settings are set for a single branchpoint). I think this holds true for the UE1 analysis in the first result. The ureteric bud sprouts from the nephric duct, from there the bud drives growth of the ureteric collecting duct without contribution from the nephric duct. The “ureteric tree” or collecting duct network is the result of migration of UB tip cells through to trunk and finally the branches as growth continues (later, in situ proliferation is a driving force for tubular extension as well). The authors, using Monocle 2 (now deprecated & outdated), describe a trajectory beginning with nephric duct, which forks into a ureteric bud fate, and a distinct ureteric trunk fate. However, Nephric duct first forms ureteric bud, and the bud differentiates into ureteric trunk (future collecting duct). This differentiation mechanism is biologically established as a linear trajectory using UB cell tracking and multiple fate analysis experiments.

Response :

Thanks for the encouraging and constructive comments. We agree that nephric duct is dispensable for the following budding and branching morphogenesis, and we have re-analyzed the UE trajectory by the package Sandrine Dudoit's Slingshot as suggested. As shown in **Extended Fig. 2d** in the revised manuscript, cells were arranged in a linear trajectory from ureteric bud/tip to ureteric trunk with ND removal, which is consistent with our knowledge of kidney development.

There needs to be a better justification for the clustering performed in the “Heterogeneity of nephrogenic progenitors” section (line 171). Subclustering the NPC cluster into eight clusters is really pushing too far given the the markers identified to justify each cluster (Ext Fig 3b). Now that we have alternative modalities to do site-specific gene expression analysis using 10X spatial transcriptomics (for example), this manuscript should use that approach in order to justify this degree of clustering. Can the authors include classical NPC markers in the plot in Ext Fig 3b? One of the clusters seems to be highly proliferative, as evidenced by the top expressed genes in Ext Fig 3b, and this might agree with Ext Fig 1b. A DimPlot with the cell cycle associations as labels is required for the subclustered NPC pool (Fig1h maybe?), this will help delineate what the different NPC clusters are doing. The “distal” NPC pool might actually be classical “metanephric mesenchyme” or semi-derived intermediate mesoderm (which is actually shown in Figure 3e) – can some markers for those cell types also be provided in Ext Fig 3b?

The pseudotime trajectory analysis with Monocle 2 is troublesome with the NPCs as well. In Fig 1J, the ***“distal” or “transition” cells differentiate into two main pools – the “differentiated” and “undifferentiated”. But this lacks biological evidence.*** What we know about NPC commitment, is that a slow cycling NPC pool typically where in Fig 1i the “CM/OriNPC” cells are. These cells then become committed (where the indNPCs are) before transitioning below the tip and forming pre-tubular aggregates soon after (where the diffNPC cells are). The “distal” and “transition” cells are perhaps metanephric mesenchyme or intermediate mesoderm at this E10.5/E11.5 age, but the ***trajectory analysis having them directly contribute in almost equal proportions to both committed and non-committed NPC fates (Fig1j) is hard to believe.***

Thanks for the insightful comments. We agree that over-clustering is a problem in this analysis, and we have revised the sub-clustering in NPC lineage and plotted classical markers, as shown in **Figure 1d**. We also re-analyzed the NPCs with package Sandrine Dudoit’s Slingshot as suggested. As shown in **Extended Figure 2e**, cells were ordered in a trajectory from IntM to MM followed by CapM, which is committed to the following nephrogenesis.

In the “Analysis of the cross talk” section, on line 227 the authors state “GDNF stood out as the only ligand produced from NPCs and received by UTip cells (Fig 2a)”. ***This is patently untrue, the figure shows a number of other ligands being upregulated in both (IL2, FGF, PTN). Regardless, the ongoing investigation using GDNF is still justified and yields***

interesting results. The authors could simply reword “stood out” to “was the most significant” (as an example).

Thanks for this comment. We have revised the introduction of GDNF in the revised manuscript.

The section on the *Ism1* conditional knockout is great, except for a statement where the authors state on line 270/271/272 “Based on the data from 170 *Ism1*^{-/-} embryos collected between E14.5 and P0, approximately 60% of homozygous mutants exhibited various kidney defects”. Followed by, on line 274, “These results demonstrated that *Ism1* is essential for kidney development”. How is that so, when 40% of the homozygous embryos collected between E14.5 and P0 had no phenotype? ***How can a kidney form normally in 40% of the embryos/pups that have conditionally removed Ism1? Is alternative splicing an issues such that the KO region is spliced out? Or is there an issue with recombination efficiency? Or functional redundancy?*** I feel that this must be addressed & explained if the authors are going to state that it is “essential” for kidney development. The authors bring this up in the discussion briefly, but the statements in the results need to be tempered down. The authors also say on line 546 in the ***Discussion “the penetration was even higher (more than 80%) when examined at E11.5”. But that is not in the results section. I don’t understand how the authors’ Ism1 mice can have 80% penetrance at E11.5, and only 60% penetrance between E14.5 and P0.***

In this same result, the authors explain that *Ism1* deficient kidney rudiments fail to enter the metanephric mesenchyme and no “t” stage is reached. This is discussed in a way that it makes the reader think that this failure is a standard feature of *Ism1* loss. However, in the phenotype we are shown that - while there is indeed agenesis - there is also kidney hypoplasia where that first budding stage has occurred, but not been particularly successful. Further, there is *unilateral* agenesis – so if an absent kidney has a contralateral kidney that is “normal”, why isn’t *Ism1* loss having the same effect there? ***This comes back to the matter of whether Ism1 can be considered “essential” for kidney development.*** The phenotype is no doubt striking, very impressive indeed, but I think it must be described in different terms.

Thanks for your insightful comments. For kidney development, around 70 genes have been reported to be linked to kidney defects, ranging from hypoplasia/dysplasia, unilateral and bilateral renal agenesis (RA). The highest penetration of RA was observed in *Ret* deficiency followed by *Gdnf* deficiency (~ 90%) and *Gfra1* deficiency (62.5%). Deletion of either the upstream mediators (*Eya1*, *Pax2*, *Six 1,2*, *Gdf11*, *Npnt* and $\alpha 8\beta 1$ -integrin) or the downstream effectors (*Etv4* and *Etv5*) of *Gdnf/Ret* pathway, also leads to RA, but with less extent of

penetration (54% in *Itga8* deficiency and 58% in *Npnt* deficiency). Thus, it is possible that loss of *Ism1* results in renal agenesis with ~60% penetration. This was possibly due to the functional redundancy. We revised the manuscript and replaced “essential” by “required” to tone down.

As to the higher penetration at E11.5 comparing to that between E14.5 and P0, it is possibly due to the existence of delayed normal development. As shown in **Extended Fig.12a**, five different types of UB morphology could be observed in the mutant at E11.5. Type I represents normal UB branching which account for about 20%. Among the 80% UB without branching structure, 20% were type II UB which was also appeared in wild-type and heterozygous embryos. Type II represents those with delayed development and could further developed into normal kidney. These percentage coincides well with the observed percentage of defective branching morphogenesis at E11.5 (~80%) and defective kidney development at birth (~60%).

Figure 4b – the WT dots on the plot are obfuscating the *Ism1*^{-/-} dots; so it is hard to compare the relative distribution of the two.

Thanks for your comment. We have quantified the percentage of each sub-cluster in the revised manuscript (**Extended Fig.6b**).

Extended Data Fig6b, 6d, 6e. Should split the plots by genotype.. i.e. [split.by = “Genotype”] in Seurat. **Then the Cell Cycle, *Ube2c* can be compared between the WT & Mut phenotypes**, and the **Dimplot in Ext 6b can be better compared between the two genotypes (the pink and blue dots overlaid can erroneously imply an abundance of one cell type from one genotype over another)**. Splitting by genotype will fix that impression.

Thanks for your comment. We have split the plots by genotype in the revised manuscript (**Extended Fig.5d-5g**).

Line 330 – what do the authors mean by “Concatenating the datasets from two genotypes at two time points”?

It means we combine the data from E10.5 and E11.5, from both WT and *Ism1*^{-/-} samples.

Line 337 onwards is poorly explained.

Line 340 – what do the authors mean by “new trunk” and “new tip” cells? How are the authors classifying these cells? **What is the difference between “tip” and “bud”; they both are *Ret*⁺, *Gfra*⁺**. The authors **haven’t shown the subclusters, and in the subclustering, how are the authors delineating “BudTip”, “PreBud”, “PreTip”, “Trunk” and “New Trunk”,**

*in the Seurat object the authors call them “cluster_name”? What representative genes are the authors relying on to classify these cells as belonging to these structures? I think this work needs to be redone using spatial transcriptomics, only then can the authors classify “Prebud” from “BudTip” (Fig 4c). What does “Pre bud” even mean? If the authors mean the first bud from the nephric duct, then that isn’t present in the E11.5 kidney. Figure 4 only represents the E11.5 data. Like before, **the Fig4B plot needs to be split by genotype – overlaying the blue and pink dots is insufficient** if the reader is required to use it as evidence for a difference in distribution across the pseudotime calculated by Monocle 2.0.*

On the UE1 pseudotime in Figure 4, Monocle 2.0 seems to be doing something odd. Transcriptionally, **“New Tip” (Fig4d) would have almost the same developmental plasticity and identity as the “bud” (4d) – however, it has placed the “new tip” population at the end of pseudotime. Why?** The authors seem to agree with what I say here when they state on line 343 - “the identity of new tips cells closely resembled that of bud cells”. Of course, they do. Monocle is again used in Fig5a, and the previous comments in this review which question the trajectory calculated by Monocle 2.0 for the NPCs (Fig1j) still stand in this instance. **In Figure 5a, the trajectory work is probably dispensable for the manuscript as it doesn’t add any significance to the work (in the way it is currently presented, at least).**

Thanks for your constructive comment. We agree that sub-clustering in a detailed manner should be supported by spatial transcriptomics. We revised the previous Figure 4 and Figure 5, and deleted the pseudotime analyzed by Monocle2 in both UE and NPC lineages. In the **revised Fig.3**, we showed that cells in ureteric epithelium failed to generate Ret signaling gradients, thus cannot form the “T-shape” structure with defined tip and trunk region. In the **revised Fig.4**, we focused more on the DEGs between genotypes and the validation of these genes.

A better description of how the bioinformatic integration was done to combine and then analyse E10.5 and E11.5 datasets is required. Were all the cells just processed as one dataset, or was each age independently classified, and then a label transfer-based integration performed?

All the cells from different sample were analyzed as one dataset, and Harmony was applied to integrate different datasets.

In my opinion, the authors need to revisit the pseudotime analysis, perhaps using an

alternative package such as **Sandrine Dudoit's Slingshot to support the trajectories generated by Monocle 2**. At the moment, the trajectories in the manuscript do not represent the biology known about the differentiation processes being described.

While the **trajectory analysis and potential overclustering** is problematic, the differential gene expression analysis is fine and the application of CellChat yields rich data which is put to good use biologically. From here on, the paper is great and insightful and enjoyable to read. Experiments are well justified, and the conclusions well supported.

Thanks for your positive comment and constructive suggestions in above. We have used the Sandrine Dudoit's Slingshot, as suggested, to re-analyze the data, in both UE and NPC lineages which is now in the **Extended Fig. 2d, 2e**.

REVIEWER COMMENTS

Reviewer #1 (Remarks to the Author):

The authors have taken on board the comments from the reviewers and improved the manuscript. Unfortunately, as the reviewer copy is not marked up, the less time-consuming job of examining revisions directly is not straightforward. In the authors responses, the authors present some alternative arguments to some concerns raised by the reviewer. This is reasonable, but if there is ambiguity around an observation, the authors should directly indicate alternative possibilities and weakness in data in drawing a conclusion. Not all data is equally strong, and accordingly, providing a perspective to interpretation does a good service to the reader and is consistent with the trend now to point to limitations in a studies conclusions.

The more appropriate paper first documenting Shh expression in detail is this one

<https://doi.org/10.1242/dev.129.22.5301>

Reviewer #2 (Remarks to the Author):

The paper is much improved.

1. The recombination experiments performed are important and do not seem inconsistent with the authors view. However, the robustness of UB growth and recombination are not easy to judge from the photos. Some of the most optimal isolated UB and recombination experiments can be found in PMID: 10377414. These depend not only on gdnf but, crucially, a metanephric mesenchymal cell condition medium which has since been shown to contain many branching factors. It is not a problem if the recombinations here fall short of this standard, but by comparing to it, the authors will be putting isthmin in the context of other factors involved in UB branching. The authors should also point to the differences in growth factor/matrix conditions and embryonic day. Factors involved in UB branching are a complex story--an important modulatory role seems fine.

2. Is the isolated UB and recombination description and protocol citation in Methods?

3. It seems some of the interpretation on the receptor end depends on b1 integrin KO in the kidney from ref 46. However, in PMID: 19439520, a similar set of experiments was done which revealed a somewhat different ureteric bud phenotype as well as a collecting system differentiation defect. Both results may be consistent with the authors view, but it is hard to tell as there is vagueness in the text. In general, the authors can better integrate the info on prior knockouts to support or modify their interpretation on the receptor end to arrive at a nuanced view.

Reviewer #3 (Remarks to the Author):

The authors have addressed most of the comments to a satisfactory level. I still have an issue with the description of Ism1 being described as "required" for kidney development (down from "essential"). I am well aware of the penetrance of the phenotype of many of the tried and tested classical markers such as Ret/Gdnf, etc. As you appreciate in your comment, up and downstream Ret *mediators* often have reduced phenotypic penetrance. Ism1 is another *mediator* of the Ret pathway, and I think that is the way Ism1 should be described.

The authors did not finish addressing Reviewer 1's comments. The rebuttal trails off at the

Supplementary (13) rebuttal.

Extended Data 2d needs cluster labels like 2e.

The co-culture of *Ism1*^{-/-} MM with *Ism1*^{+/+} WT UB & *Ism1*^{+/+} MM with *Ism1*^{-/-} UB in matrigel is a striking result. Even with exogenous Gdnf in the matrigel, it implies *Ism1* deficient MM as the primary effector in the phenotype.

Point to Point response to reviewers

Reviewer #1 (Remarks to the Author):

The authors have taken on board the comments from the reviewers and improved the manuscript. Unfortunately, as the reviewer copy is not marked up, the less time-consuming job of examining revisions directly is not straightforward. In the authors responses, the authors present some alternative arguments to some concerns raised by the reviewer. This is reasonable, but if there is ambiguity around an observation, the authors should directly indicate alternative possibilities and weakness in data in drawing a conclusion. Not all data is equally strong, and accordingly, providing a perspective to interpretation does a good service to the reader and is consistent with the trend now to point to limitations in a studies conclusions.

We are sorry for the inconvenience caused. We have now included track changes in the revised manuscript.

We thank the advice on the data presentation and we now included alternative explanations to some of the observations. We provided a perspective to the limitation of current work and ways to address those issues (Paragraph 3, 4, 6 in DISCUSSION).

The more appropriate paper first documenting Shh expression in detail is this one <https://doi.org/10.1242/dev.129.22.5301>

Thanks for the advice and reference. We have modified the description of *Shh*⁺ cell cluster and its citations (**Ref 29**) in the revised manuscript.

Reviewer #2 (Remarks to the Author):

The paper is much improved.

1. The recombination experiments performed are important and do not seem inconsistent with the authors view. However, the robustness of UB growth and recombination are not easy to judge from the photos. Some of the most optimal isolated UB and recombination experiments can be found in PMID: 10377414. These depend not only on *gdnf* but, crucially, a metanephric mesenchymal cell condition medium which has since been shown to contain many branching factors. It is not a problem if the recombinations here fall short of this standard, but by comparing to it, the authors will be putting *isthmin* in the context of other factors involved in UB branching. The authors should also point to the differences in growth factor/matrix conditions and embryonic day. Factors involved in UB branching are a complex story--an important modulatory role seems fine.

Thanks for the advice and reference. We now added the recombination experiments in the revised manuscript (**Extended Figure12 c-e, Ref 62**) to support the modulatory role of *Isml* in UB branching morphogenesis.

2. Is the isolated UB and recombination description and protocol citation in Methods?

We now added the data of recombination experiments in the extended Figure 12, we have incorporated the description and reference in Methods (**Ref 62**).

3. It seems some of the interpretation on the receptor end depends on $\beta 1$ integrin KO in the kidney from ref 46. However, in PMID: 19439520, a similar set of experiments was done which revealed a somewhat different ureteric bud phenotype as well as a collecting system differentiation defect. Both results may be consistent with the authors view, but it is hard to tell as there is vagueness in the text. In general, the authors can better integrate the info on prior knockouts to support or modify their interpretation on the receptor end to arrive at a nuanced view.

Thanks for the advice. We have revised the discussion on $\beta 1$ integrin-related ureteric bud phenotypes. (**Paragraph 3 of Discussion part, Ref 48**).

Reviewer #3 (Remarks to the Author):

The authors have addressed most of the comments to a satisfactory level. I still have an issue with the description of *Ism1* being described as "required" for kidney development (down from "essential"). I am well aware of the penetrance of the phenotype of many of the tried and tested classical markers such as *Ret*/*Gdnf*, etc. As you appreciate in your comment, up and downstream *Ret* *mediators* often have reduced phenotypic penetrance. *Ism1* is another *mediator* of the *Ret* pathway, and I think that is the way *Ism1* should be described.

Thanks for the comment. We have revised the manuscript and replaced "required" with "modulate or mediate".

The authors did not finish addressing Reviewer 1's comments. The rebuttal trails off at the Supplementary (13) rebuttal.

Thanks for your careful observation. That comment (13) was not fully shown in the first round reviewer 1's comments. It is likely deleted by the reviewer 1.

Extended Data 2d needs cluster labels like 2e.

We have added in the revised manuscript (**Extended Data 2d**).

The co-culture of *Ism1*^{-/-} MM with *Ism1*^{+/+} WT UB & *Ism1*^{+/+} MM with *Ism1*^{-/-} UB in matrigel is a striking result. Even with exogenous *Gdnf* in the matrigel, it implies *Ism1* deficient MM as the primary effector in the phenotype.

Thanks for your comments. The co-culture result was added in the revised manuscript in **Extended Figure12** to support the role of *Ism1* in UB branching morphogenesis.

We thank the reviewer's comment which is in agreement of our conclusion. Regulation of the renal branching morphogenesis by *Ism1* is MM-dependent.

REVIEWERS' COMMENTS

Reviewer #2 (Remarks to the Author):

no further comments

Reviewer #3 (Remarks to the Author):

I'm satisfied with the amendments except for the speculation from 468 onwards where the authors discuss the "slim chance" for $\alpha 3\beta 1$ or $\alpha 6\beta 1$ to act as receptors for Ism1. That may be true, but it is subjective/speculative and should be reworded/rephrased.

The manuscript still requires some minor grammar corrections.

e.g. line 427

"null mice exhibited reduced number of collecting ducts and abnormal glomerulus in"
should read

"null mice exhibited a reduced number of collecting ducts and abnormal glomeruli in"

another example from line 464:

"Selective deletion of integrin $\alpha 3$ in UB results in abnormal or absent of the kidney papillae"
should read

"Selective deletion of integrin $\alpha 3$ in UB results in abnormal or absent kidney papillae"

There are multiple other similar grammatical changes required.

Line 464 (the UB), 465 (led to a medullary defect), and others.

Point-to-Point response to the reviewers

Reviewer #2 (Remarks to the Author):

no further comments.

Reviewer #3 (Remarks to the Author):

I'm satisfied with the amendments except for the speculation from 468 onwards where the authors discuss the "slim chance" for $\alpha3\beta1$ or $\alpha6\beta1$ to act as receptors for Ism1 . That may be true, but it is subjective/speculative and should be reworded/rephrased.

The manuscript still requires some minor grammar corrections.

e.g. line 427

"null mice exhibited reduced number of collecting ducts and abnormal glomerulus in" should read "null mice exhibited a reduced number of collecting ducts and abnormal glomeruli in" another example from line 464:

"Selective deletion of integrin $\alpha3$ in UB results in abnormal or absent of the kidney papillae" should read "Selective deletion of integrin $\alpha3$ in UB results in abnormal or absent kidney papillae"

There are multiple other similar grammatical changes required.

Line 464 (the UB), 465 (led to a medullary defect), and others.

Response:

We thank the reviewer for the comments. Following the reviewer's advice, we have checked and corrected the English language throughout the manuscript.